# Learning Nash equilibria in Rank-1 games

**Nikolas Patris**
University of California, Irvine
Archimedes Research Unit
npatris@uci.edu

**Ioannis Panageas**
University of California, Irvine
ipanagea@ics.uci.edu

## Abstract

Learning Nash equilibria (NE) in games has garnered significant attention, particularly in the context of training Generative Adversarial Networks (GANs) and multi-agent Reinforcement Learning. The current state-of-the-art in efficiently learning NE in games focuses on landscapes that meet the (weak) minty property or games characterized by a unique function, often referred to as potential games. A significant challenge in this domain is that computing Nash equilibria is a computationally intractable task Daskalakis et al. (2009).

In this paper we focus on bimatrix games $(\mathbf{A}, \mathbf{B})$ called rank-1. These are games in which the sum of the payoff matrices $\mathbf{A} + \mathbf{B}$ is a rank 1 matrix; note that standard zero-sum games are rank-0. We show that a modification of optimistic mirror descent converges to an $\epsilon$-approximate NE after $O\left(\frac{1}{\epsilon^2}\log(\frac{1}{\epsilon})\right)$ iterates in rank-1 games. We achieve this by leveraging structural results about the NE landscape of rank-1 games Adsul et al. (2021). Notably, our approach bypasses the fact that these games do not satisfy the MVI property.

## 1 Introduction

Learning in games has its origins in Blackwell's influential work on the approachability theorem (Blackwell, 1956; Abernethy et al., 2011), which paved the way for the development of various learning algorithms with convergence guarantees to different game theoretic solution concepts, including coarse correlated equilibrium (CCE) and Nash equilibrium (NE). For instance, a well-known result in this context is that if both players in a zero-sum game employ a no-regret learning algorithm, the empirical averages of their strategies will converge to a NE. However, the typical guarantees provided by online learning frameworks do not shed light on how the system stabilizes toward a Nash equilibrium when the underlying two-player game is neither zero-sum nor a potential game (Anagnostides et al., 2022b). In such cases, one can only argue that the empirical averages converge to a CCE if both players employ a no-regret learning algorithm. This raises the following fundamental question:

*For which classes of two-player games, beyond zero-sum and potential games, can we design online learning algorithms that guarantee convergence to a Nash equilibrium?*

The work in Kannan & Theobald (2010) introduced a hierarchy of bimatrix games denoted as $(\mathbf{A}, \mathbf{B})$, where the rank of $\mathbf{A} + \mathbf{B}$ is $k$. These are referred to as rank-$k$ bimatrix games. Notably, this class encompasses zero-sum games when $k$ is zero. Intriguingly, it has been demonstrated that even for the case of $k = 1$, the set of Nash equilibria can comprise a potentially large number of disconnected components and lacks convexity. Furthermore, it has been proven that for $k \geq 3$, computing an *exact* Nash equilibria in rank $k$ bimatrix games is a PPAD-hard problem Mehta (2018). In other words, it is highly unlikely that a polynomial-time algorithm exists for computing Nash equilibria in such games. On the contrary, rank-1 games have been shown to have polynomial-time algorithms based on linear programming formulations Adsul et al. (2021; 2011), while the complexity of rank-2 games remains an open question.

**Our contributions and Technical Overview**  In this paper, we focus on rank-1 games. First we establish that there exist rank-1 games that do not satisfy the so-called Minty criterion and as a result *optimistic mirror descent* Rakhlin & Sridharan (2013) and *extra gradient* algorithms Korpelevich (1976) fail to converge to a Nash equilibrium.

On a positive note, we introduce a decentralized learning algorithm that converges to an $\epsilon$-approximate Nash equilibrium in $O\left(\frac{1}{\epsilon^2}\log\left(\frac{1}{\epsilon}\right)\right)$ iterations, as proven in Theorem 3.7. Both agents in the game utilize an optimistic mirror descent algorithm in a parametrized zero-sum game that evolves over time. The key to our technical analysis lies in how we adapt the underlying parameter, ensuring that when the algorithm terminates, an approximate Nash equilibrium in the parametrized zero-sum game coincides with an approximate Nash equilibrium in the rank-1 game. To provide more context, consider a rank-1 bimatrix game represented as $(\mathbf{A}, \mathbf{B})$. We can assume that $\mathbf{B} = -\mathbf{A} + \boldsymbol{a}\boldsymbol{b}^\top$ for some vectors $\boldsymbol{a}$ and $\boldsymbol{b}$. It turns out that a Nash equilibrium $(\boldsymbol{x}, \boldsymbol{y})$ of the rank-1 game $(\mathbf{A}, \mathbf{B})$ is also a Nash equilibrium of the parametrized zero-sum game $(\mathbf{A} - \mathbb{1}\lambda\boldsymbol{b}^\top, -\mathbf{A} + \mathbb{1}\lambda\boldsymbol{b}^\top)$ for an appropriate choice of the parameter $\lambda$, where $\mathbb{1}$ is the all-ones vector. The primary technical challenge is formulating a suitable optimization problem to ensure that a Nash equilibrium in the parameterized translates to a Nash equilibrium in the rank-1 game, as shown in Lemma 3.4.

Our proposed algorithm is based on *optimistic mirror descent* applied to the parameterized bilinear function $\boldsymbol{x}^\top(\mathbf{A} - \mathbb{1}\lambda\boldsymbol{b}^\top)\boldsymbol{y}$ with an additional regularization term $(\boldsymbol{x}^\top\boldsymbol{a} - \lambda)^2$. Depending on the sign of the difference $(\boldsymbol{x}^\top\boldsymbol{a} - \lambda)$, $\lambda$ is updated appropriately. To prove our claims, we leverage the structural insights of the Nash equilibrium landscape Adsul et al. (2021) and exploit the stationarity properties of the $\lambda$-parameterized (strongly) concave-convex function $f_\lambda(\boldsymbol{s}, \boldsymbol{y}) = \boldsymbol{x}^\top(\mathbf{A} - \mathbb{1}\lambda\boldsymbol{b}^\top)\boldsymbol{y} - \frac{1}{2}(\boldsymbol{x}^\top\boldsymbol{a} - \lambda)^2$. Furthermore, in Appendix C we provide experimental evaluation supporting our theoretical findings.

## 2  PRELIMINARIES

### 2.1  NOTATION AND DEFINITIONS

**Notation**  Let $\mathbb{R}$ be the set of real numbers, and $[n]$ be the set $\{1, 2, \ldots, n\}$. We define $\Delta_n$ as the probability simplex, which is the set of $n$-dimensional probability vectors, i.e., $\Delta_n := \left\{\boldsymbol{x} \in \mathbb{R}^n : \boldsymbol{x}_i \geq 0, \sum_{i=1}^n \boldsymbol{x}_i = 1\right\}$. We use $\boldsymbol{e}_i$ to denote the $i$-th elementary vector, and to refer to the $i$-th coordinate of $\boldsymbol{x}$ we use $\boldsymbol{x}_i$. The superscripts are used to indicate the timestep at which a vector is referring to.

**Normal-form Games**  We primarily focus on *two-player* normal-form bimatrix game. We are given a pair of payoff matrices $\mathbf{A} \in \mathbb{R}^{n \times m}$ and $\mathbf{B} \in \mathbb{R}^{n \times m}$ where $n$ and $m$ are the respective pure strategies (actions) of the *row* and the *column* player. After the players choose their action, we denote $(\boldsymbol{e}_i, \boldsymbol{e}_j)$ the joint action profile for $i \in [n]$ and $j \in [m]$. Then, their respective payoffs are $\mathbf{A}_{ij} := \boldsymbol{e}_i^\top \mathbf{A}\boldsymbol{e}_j$ for the row player and $\mathbf{B}_{ij} := \boldsymbol{e}_i^\top \mathbf{B}\boldsymbol{e}_j$ for the column player. Player are also allowed to randomize. A *mixed strategy* for the row (resp. column) player is a probability distribution $\boldsymbol{x} \in \Delta_n$ (resp. $\boldsymbol{y} \in \Delta_m$) over the $n$ rows (resp. $m$ columns). In this case, the *expected utility* of the row, column player is expressed as $\boldsymbol{x}^\top \mathbf{A}\boldsymbol{y}$ and $\boldsymbol{x}^\top \mathbf{B}\boldsymbol{y}$ respectively, where $(\boldsymbol{x}, \boldsymbol{y}) \in \Delta_n \times \Delta_m$ denotes the joint mixed strategy profile.

**Definition 2.1** ($\epsilon$-approximate Nash Equilibrium). A strategy profile $(\boldsymbol{x}^\star, \boldsymbol{y}^\star) \in \Delta_n \times \Delta_m$ is called an $\epsilon$-approximate Nash equilibrium (NE) if and only if

$$(\boldsymbol{x}^\star)^\top\mathbf{A}\boldsymbol{y}^\star \geq \boldsymbol{x}^\top\mathbf{A}\boldsymbol{y}^\star - \epsilon \ \ \forall \boldsymbol{x} \in \Delta_n \quad \text{and} \quad (\boldsymbol{x}^\star)^\top\mathbf{B}\boldsymbol{y}^\star \geq (\boldsymbol{x}^\star)^\top\mathbf{B}\boldsymbol{y} - \epsilon \ \ \forall \boldsymbol{y} \in \Delta_m$$

In words, an approximate Nash equilibrium is a strategy profile in which no player can improve their payoff significantly (more than $\epsilon$) by unilaterally changing their strategy, but the strategy profile may not necessarily satisfy the definition of an exact ($\epsilon = 0$) Nash equilibrium.

**Remark 2.2.** We highlight two special cases of an $\epsilon$-approximate Nash equilibrium. Firstly, when $\epsilon$ is equal to zero, it is referred to as an *exact* Nash equilibrium. Secondly, when the *support* of strategies, $\text{supp}(\boldsymbol{x}) = \{i \mid \boldsymbol{x}_i > 0\}$, is of size 1, it is called a *pure* Nash equilibrium.

## 2.2 RANK-1 GAMES

In this work, we consider bimatrix games $(\mathbf{A}, \mathbf{B})$ of fixed rank, and particularly of rank 1. This hierarchy within two-player games was initially introduced by Kannan & Theobald (2010) in an attempt to offer a more detailed characterization of games that (may) admit simple polynomial-time algorithms based on the rank of the matrix $(\mathbf{A} + \mathbf{B})$. In a related model, Lipton et al. (2003) investigated games where *both* payoff matrices are of fixed rank $k$, and they proposed a simple (quasi) polynomial-time algorithm for constructing an $\epsilon$-approximate NE. This seemingly similar model, however, has a significant drawback; not even zero-sum games, which have polynomial-time algorithms, entirely belong to this class. Motivated by this observation, Kannan & Theobald (2010) proposed that $(\mathbf{A} + \mathbf{B})$ should have fixed rank instead.

**Definition 2.3** (Games of fixed rank). The rank of a bimatrix game $(\mathbf{A}, \mathbf{B})$ is the rank of matrix $\mathbf{A} + \mathbf{B}$.

**Remark 2.4.** It is apparent that zero-sum games $(\mathbf{A}, -\mathbf{A})$ are rank-0 games as $\mathbf{A} + (-\mathbf{A}) = \mathbf{0}$. Additionally, rank-1 games satisfy $\mathbf{A} + \mathbf{B} = \boldsymbol{a}\boldsymbol{b}^\top$.

**Assumption 2.5.** As Nash equilibria remain invariant under the transformations of shifting and scaling of the payoff matrices, we assume that the elements of $\boldsymbol{a}$ and $\boldsymbol{b}$ are confined within the interval $[-1, 1]$. This assumption simplifies our calculations without impacting any of our results.

The following lemma presents an alternative method for describing a bimatrix game of fixed rank $r$.

**Lemma 2.6** (Lemma 5 Adsul et al. (2021)). *An $(n \times m)$ bimatrix game $(\mathbf{A}, \mathbf{B})$ of positive rank $r$ can be written as $(\mathbf{A}, \mathbf{C} + \boldsymbol{a}\boldsymbol{b}^\top)$ for suitable $\boldsymbol{a} \in \mathbb{R}^n, \boldsymbol{b} \in \mathbb{R}^m$, and a game $(\mathbf{A}, \mathbf{C})$ of rank $r - 1$.*

The following lemma, though simple, holds significant importance as it establishes that for a bimatrix game of fixed rank $r$, the set of its Nash equilibria is equivalent to the intersection of the set of equilibria of a parameterized game of rank $r - 1$ and a hyperplane.

**Lemma 2.7** (Lemma 6 Adsul et al. (2021)). *Let $\mathbf{A}, \mathbf{C} \in \mathbb{R}^{n \times m}$, $\boldsymbol{x} \in \Delta_n$, $\boldsymbol{y} \in \Delta_m$, $\boldsymbol{a} \in \mathbb{R}^n$, $\boldsymbol{b} \in \mathbb{R}^m$, $\lambda \in \mathbb{R}$. The following are equivalent:*

*(a) $(\boldsymbol{x}, \boldsymbol{y})$ is an equilibrium of $(\mathbf{A}, \mathbf{C} + \boldsymbol{a}\boldsymbol{b}^\top)$.*

*(b) $(\boldsymbol{x}, \boldsymbol{y})$ is an equilibrium of $(\mathbf{A}, \mathbf{C} + \mathbb{1}\lambda\boldsymbol{b}^\top)$ and $\boldsymbol{x}^\top\boldsymbol{a} = \lambda$.*

*(c) $(\boldsymbol{x}, \boldsymbol{y})$ is an equilibrium of $(\mathbf{A} - \mathbb{1}\lambda\boldsymbol{b}^\top, \mathbf{C} + \mathbb{1}\lambda\boldsymbol{b}^\top)$ and $\boldsymbol{x}^\top\boldsymbol{a} = \lambda$.*

*Sketch Proof.* The equivalence of (a) and (b) holds because the players get in both games the same expected payoffs for their pure strategies. The games in (b) and (c) are strategically equivalent. $\square$

While it may not be immediately evident how this rank reduction can be practically applied, the true significance of Lemma 2.7 becomes fully apparent in the context of rank-1 games.

**Corollary 2.8.** *Let $(\mathbf{A}, \mathbf{B})$ be a bimatrix game of rank 1, i.e. $\mathbf{A} + \mathbf{B} = \boldsymbol{a}\boldsymbol{b}^\top$. Then, the following are equivalent.*

*(a) $(\boldsymbol{x}, \boldsymbol{y})$ is an equilibrium of $(\mathbf{A}, -\mathbf{A} + \boldsymbol{a}\boldsymbol{b}^\top)$.*

*(b) $(\boldsymbol{x}, \boldsymbol{y})$ is an equilibrium of $(\mathbf{A} - \mathbb{1}\lambda\boldsymbol{b}^\top, -\mathbf{A} + \mathbb{1}\lambda\boldsymbol{b}^\top)$ and $\boldsymbol{x}^\top\boldsymbol{a} = \lambda$.*

The proof follows trivially from the equivalence of (a) and (c) in Lemma 2.7.

**Parameterized zero-sum games** Corollary 2.8 establishes a connection between rank-1 games and parameterized zero-sum games. While this link initially appears compelling, it remains uncertain whether an algorithm can *compute*, or more importantly *learn*, a Nash equilibrium $(\boldsymbol{x}, \boldsymbol{y})$ satisfying $\boldsymbol{x}^\top \boldsymbol{a} = \lambda$ at the same time. The primary challenge arises from the fact that optimal strategies vary as the parameter $\lambda$ changes. It is well-known, however, that zero-sum games can be efficiently solved using various methods, including LP-based techniques Adler (2013); von Stengel (2023) and no-regret algorithms Freund & Schapire (1999); Daskalakis et al. (2011); Syrgkanis et al. (2015); Chen & Peng (2020). Therefore, as long as there is an easy way of updating $\lambda$, the parameterized zero-sum game serves as an *intermediary* to solve the rank-1 game.

An initial attempt was made in Adsul et al. (2011), where they demonstrated that the set of fully-labeled points, which captures all Nash equilibria of the game, of specific best-response polytopes contains only a path that exhibits monotonicity with respect to $\lambda$. Additionally, they introduced an algorithm that employs binary search on the parameter to compute an *exact* Nash equilibrium for the rank-1 game. However, a limitation of this method is its inability to handle non-degenerate games. In their subsequent work Adsul et al. (2021), while retaining their structural insights and building on Adler & Monteiro (1992), they refined their approach. They proposed a polynomial-time algorithm, similar in spirit to the one in Adsul et al. (2011), to compute a Nash equilibrium for *any* rank-1 game.

**Definition 2.9.** Let $\mathcal{N}$ be the set of the Nash equilibria of the parameterized zero-sum game defined as follows.

$$\mathcal{N} := \{(\lambda, \boldsymbol{x}, \boldsymbol{y}) \in \mathbb{R} \times \mathbb{R}^n \times \mathbb{R}^m \,|\, (\boldsymbol{x}, \boldsymbol{y}) \text{ is a NE of } \left(\mathbf{A} - \mathbb{1}\lambda\boldsymbol{b}^\top, -\mathbf{A} + \mathbb{1}\lambda\boldsymbol{b}^\top\right)\} \tag{1}$$

The following key lemma, essential for the algorithm in Adsul et al. (2021), shows that the set $\mathcal{N}$ exhibits monotonicity with respect to $\lambda$.

**Lemma 2.10** (Lemma 17 Adsul et al. (2021)). *Let* $\underline{\lambda} \le \overline{\lambda}$, $\underline{\boldsymbol{x}}, \overline{\boldsymbol{x}} \in \Delta_n$ *and* $\underline{\boldsymbol{y}}, \overline{\boldsymbol{y}} \in \Delta_m$ *so that for* $\mathcal{N}$ *in* (1)

$$\left(\underline{\lambda}, \underline{\boldsymbol{x}}, \underline{\boldsymbol{y}}\right) \in \mathcal{N}, \quad \underline{\lambda} \le \underline{\boldsymbol{x}}^\top \boldsymbol{a}, \quad \left(\overline{\lambda}, \overline{\boldsymbol{x}}, \overline{\boldsymbol{y}}\right) \in \mathcal{N}, \quad \overline{\lambda} \ge \overline{\boldsymbol{x}}^\top \boldsymbol{a}.$$

*Then* $\boldsymbol{x}^\top \boldsymbol{a} = \lambda$ *for some* $(\lambda, \boldsymbol{x}, \boldsymbol{y}) \in \mathcal{N}$ *with* $\lambda \in [\underline{\lambda}, \overline{\lambda}]$.

This property serves as the invariant maintained by the algorithm outlined in Adsul et al. (2021). The algorithm essentially employs a binary search with respect to the parameter $\lambda$. In brief, at each iteration, it considers the midpoint, $\lambda = (\underline{\lambda} + \overline{\lambda})/2$, and, based on its value, it solves a series of linear programs. These either return a Nash equilibrium of the zero-sum game $\left(\mathbf{A} - \mathbb{1}\lambda\boldsymbol{b}^\top, -\mathbf{A} + \mathbb{1}\lambda\boldsymbol{b}^\top\right)$ for which $\boldsymbol{x}^\top \boldsymbol{a} = \lambda$ or indicate that none of the Nash equilibrium strategies satisfy this equation. Importantly, in the latter case, and due to the monotonic nature with respect to the parameter, the sign of the difference $(\boldsymbol{x}^\top \boldsymbol{a} - \lambda)$ should be maintained. Therefore, depending on it, either $\underline{\lambda}$ or $\overline{\lambda}$ is updated accordingly. For instance, in case $\boldsymbol{x}^\top \boldsymbol{a} - \lambda < 0$ then $\underline{\lambda} = \lambda$; otherwise, it sets $\overline{\lambda} = \lambda$ [1]. According to Lemma 2.10, this step is safe as there exists a Nash equilibrium $(\boldsymbol{x}, \boldsymbol{y})$ such that $\boldsymbol{x}^\top \boldsymbol{a} = \lambda$ and $\lambda \in [\underline{\lambda}, \overline{\lambda}]$. In Lemma 3.5, we extend this property to the case of an *approximate* Nash equilibrium by leveraging the convexity of Nash equilibria in zero-sum games.

## 2.3 BASIC BACKGROUND ON MIN-MAX OPTIMIZATION

After establishing the link between rank-1 games and zero-sum games, as demonstrated in Corollary 2.8, this section presents a brief background on min-max optimization. A more compact and thorough presentation is deferred to Appendix B.

---

[1] We have intentionally omitted certain technical details of this step that do not impact the core aspects of the algorithm.

A Nash equilibrium in zero-sum games represents a solution to a saddle point problem which in its general form (not only for bilinear functions) is

$$\min_{\boldsymbol{x}\in\mathcal{X}} \max_{\boldsymbol{y}\in\mathcal{Y}} f(\boldsymbol{x}, \boldsymbol{y}) \tag{2}$$

where $\mathcal{X}, \mathcal{Y}$ are convex and compact subsets of an Euclidean $n$-dimensional ($m$-dimensional resp.) space, and $f$ is a continuously differentiable function, i.e. a smooth function.

**Remark 2.11.** It is widely recognized that two zero-sum games constitute a special case where the function $f(\boldsymbol{x}, \boldsymbol{y}) = \boldsymbol{x}^\top \mathbf{A} \boldsymbol{y}$ is convex (linear) with respect to $\boldsymbol{x}$ and concave (linear) with respect to $\boldsymbol{y}$. Then, the celebrated minimax theorem v. Neumann (1928) asserts that $\min_{\boldsymbol{x}\in\mathcal{Y}} \max_{\boldsymbol{y}\in\mathcal{Y}} f(\boldsymbol{x}, \boldsymbol{y}) = \max_{\boldsymbol{y}\in\mathcal{Y}} \min_{\boldsymbol{x}\in\mathcal{X}} f(\boldsymbol{x}, \boldsymbol{y})$.

However, in practical scenarios, such as those in Generative Adversarial Networks (GANs) Daskalakis et al. (2017), the function $f$ does not meet the necessary convex-concave property, and so most optimization techniques for (2) do not to apply. As a result, a significant amount of research has focused instead on different notions of local min-max solutions Daskalakis & Panageas (2018b); Jin et al. (2020). The result in Daskalakis et al. (2021), however, establishes that for general smooth objectives, the computation of even approximate first-order locally optimal min-max solutions is intractable. In light of this intractability result, research has shifted its focus to investigate structural properties that circumvent this barrier, enabling the efficient computation of a solution. This will usually involve imposing some (mild or not) assumptions on the objective function. An example of such an assumption is the existence of a solution to (MVI), as presented below.

**Nash Gap**    The goal is to find a point $(\boldsymbol{x}^\star, \boldsymbol{y}^\star) \in \mathcal{X} \times \mathcal{Y}$ that is *close* to the set of Nash equilibria. The notion of closeness is implied in terms of the best response gap, not the distance to the set of NE [2], and is defined with respect to the (Nash) gap as

$$\mathrm{Gap}(\boldsymbol{x}^\star, \boldsymbol{y}^\star) := \max_{\boldsymbol{y}\in\mathcal{Y}} f(\boldsymbol{x}^\star, \boldsymbol{y}) - \min_{\boldsymbol{x}\in\mathcal{X}} f(\boldsymbol{x}, \boldsymbol{y}^\star), \tag{Gap}$$

which is always non-negative and zero if and only if $(\boldsymbol{x}^\star, \boldsymbol{y}^\star)$ is a NE.

**Variational inequalities**    We introduce the set $\mathcal{Z} = \mathcal{X} \times \mathcal{Y}$ and the operator $F$ associated with the function $f$, defined as $F(\boldsymbol{z}) = F(\left[\begin{smallmatrix}\boldsymbol{x}\\\boldsymbol{y}\end{smallmatrix}\right]) = \left[\begin{smallmatrix}\nabla_{\boldsymbol{x}} f(\boldsymbol{x}, \boldsymbol{y})\\-\nabla_{\boldsymbol{y}} f(\boldsymbol{x}, \boldsymbol{y})\end{smallmatrix}\right]$. It is readily verified that a saddle point problem (2) can be framed as a variational inequality problem; this unifying treatment enables the application of various techniques and results drawn from the vast literature of variational inequalities, Facchinei & Pang (2003). Given the compact convex set $\mathcal{Z}$ and an operator $F$, the *Stampacchia Variational Inequality* problem consists in finding $\boldsymbol{z}^\star \in \mathcal{Z}$ such that:

$$\langle F(\boldsymbol{z}^\star), \boldsymbol{z} - \boldsymbol{z}^\star \rangle \geq 0 \quad \text{for all } \boldsymbol{z} \in \mathcal{Z} \tag{SVI}$$

In this case, $\boldsymbol{z}^\star$ is referred to as the *strong* solution to variational inequality problem corresponding to $F$. From a game-theoretic standpoint, $\boldsymbol{z}^\star$ corresponds to a Nash equilibrium of the underlying game. As the existence of a Nash equilibrium is always guaranteed to exist Nash Jr (1950), we can safely assume that there exists a solution to (SVI).

Most of the existing literature on variational inequality problems Facchinei & Pang (2003) has focused on the monotone case, i.e. $f$ is convex concave. In this context, it is widely acknowledged that every solution to (SVI) is also a solution to the *Minty Variational Inequality* problem (MVI).

$$\langle F(\boldsymbol{z}), \boldsymbol{z} - \boldsymbol{z}^\star \rangle \geq 0 \quad \text{for all } \boldsymbol{z} \in \mathcal{Z} \tag{MVI}$$

---

[2]This notion of closeness is commonly referred to in the literature as weak, while the closeness in terms of the distance to a Nash Equilibrium is known as the strong notion Etessami & Yannakakis (2010).

In settings beyond the monotone case, all that can be said is that the set of solutions for (MVI) is a subset of the set of solutions for (SVI). An important (Minty) criterion that ensures tractability is the existence of a solution to (MVI). In Mertikopoulos et al. (2018), they introduced the concept of coherence (refer to Definition 2.1), which holds true if every saddle point of $f$ (2), and in turn to (SVI), is a solution to (MVI) and vice versa. In Mertikopoulos et al. (2018); Song et al. (2020); Zhou et al. (2020) (and references therein) it has been established that if the Minty criterion is satisfied (e.g. bilinear problems, quasi-convex-concave objectives) then extra gradient Korpelevich (1976) do converge to a solution of saddle point problem. In recent works, such as Diakonikolas et al. (2021); Pethick et al. (2022), a less stringent criterion, namely the weak Minty criterion, has been explored, and convergence to a solution of (2) was also ensured. Nevertheless, as we will demonstrate shortly in the Section 3.1, even relatively simple rank-1 games fail to satisfy the (weak) Minty criterion.

## 3 EFFICIENTLY LEARNING NASH EQUILIBRIA IN RANK-1 GAMES

In this section, we sketch our main findings concerning the learning of Nash equilibria in rank-1 games; all the proofs are deferred to Appendix A. We start by presenting two illustrative examples that highlight the inherent challenges of these games. Despite their apparent simplicity, conventional first-order methods fail to converge to a Nash equilibrium. In Section 3.2, we introduce an efficient algorithm for learning a NE, that uses *and* extends the structural properties detailed in Section 2.2.

### 3.1 TECHNICAL CHALLENGES

The rank-reduction method presented in Corollary 2.8 transforms a rank-1 game $(\mathbf{A}, \mathbf{B})$ into a parameterized zero-sum game $(\mathbf{A} - \mathbb{1}\lambda\boldsymbol{b}^\top, -\mathbf{A} + \mathbb{1}\lambda\boldsymbol{b}^\top)$. A question that arises is whether the set of *approximate* Nash equilibria of the zero-sum game is properly included in the *approximate* Nash equilibria of the rank-1 game. Although in the case of an *exact* NE, we readily verify that it does not hold, the former remains unclear for *approximate* NE. In other words, can solving the zero-sum game alone provide a solution for the rank-1 game?

The following example provides a negative answer to this question, even when we have complete knowledge of a correct value $\lambda^\star$-i.e., corresponding to a Nash equilibrium $(\boldsymbol{x}^\star, \boldsymbol{y}^\star)$ of the rank-1 game where $(\boldsymbol{x}^\star)^\top \boldsymbol{a} = \lambda^\star$.

**Example 3.1** (Full Version in Example A.1). Let $\mathbf{A} - \mathbb{1}\lambda\boldsymbol{b}^\top = \begin{bmatrix} 1 & 1 \\ 1-\epsilon & 1-\epsilon \end{bmatrix}$. This game has an exact NE of the form $(\boldsymbol{x}^\star, \boldsymbol{y}^\star) = ((1,0), (\mu, 1-\mu))$ for any $\mu \in [0,1]$. Additionally, $(\tilde{\boldsymbol{x}}, \tilde{\boldsymbol{y}}) = ((\frac{1}{2}, \frac{1}{2}), (\frac{1}{2}, \frac{1}{2}))$ is an $\frac{\epsilon}{2}$-approximate Nash equilibrium of the zero-sum game $(\mathbf{A} - \mathbb{1}\lambda\boldsymbol{b}^\top, -\mathbf{A} + \mathbb{1}\lambda\boldsymbol{b}^\top)$

$$\boldsymbol{x}^{\star\top} \left( \mathbf{A} - \mathbb{1}\lambda\boldsymbol{b}^\top \right) \boldsymbol{y}^\star = 1 \text{ and } \tilde{\boldsymbol{x}}^\top \left( \mathbf{A} - \mathbb{1}\lambda\boldsymbol{b}^\top \right) \tilde{\boldsymbol{y}} = \begin{bmatrix} \frac{1}{2} & \frac{1}{2} \end{bmatrix} \begin{bmatrix} 1 & 1 \\ 1-\epsilon & 1-\epsilon \end{bmatrix} \begin{bmatrix} \frac{1}{2} \\ \frac{1}{2} \end{bmatrix} = 1 - \frac{\epsilon}{2}$$

We choose $\boldsymbol{a} = \begin{bmatrix} 1 & 2 \end{bmatrix}^\top$ and $\boldsymbol{b} = \begin{bmatrix} 2 & 1 \end{bmatrix}^\top$, and so $\boldsymbol{x}^{\star\top} \boldsymbol{a} = 1 = \lambda$.

$$\mathbf{A} = \begin{bmatrix} 1 & 1 \\ 1-\epsilon & 1-\epsilon \end{bmatrix} + \begin{bmatrix} 2 & 1 \\ 2 & 1 \end{bmatrix} = \begin{bmatrix} 3 & 2 \\ 3-\epsilon & 2-\epsilon \end{bmatrix} \quad \text{and} \quad \mathbf{B} = \begin{bmatrix} -1 & -1 \\ 1+\epsilon & \epsilon \end{bmatrix}$$

We confirm that $(\tilde{\boldsymbol{x}}, \tilde{\boldsymbol{y}}) = \left( \left(\frac{1}{2}, \frac{1}{2}\right), \left(\frac{1}{2}, \frac{1}{2}\right) \right)$ no longer qualifies as an $\frac{\epsilon}{2}$-approximate Nash Equilibrium for the original rank-1 game as the Nash equilibrium condition (inequality) for the column player does not hold.

As described in Section 2.3, there are additional structural properties that enables convergence to a solution even if the game is no longer monotone. As such, our next goal is to investigate whether the Minty criterion (MVI) is satisfied rank-1 games. Unfortunately, but importantly for the scope of this work, it fails to hold.

---

**Algorithm 1:** Rank-1 Game Solver

**Input:** $\boldsymbol{x}^{(0)} = \mathbb{1}/n \in \Delta_n$, $\boldsymbol{y}^{(0)} = \mathbb{1}/m \in \Delta_m$
**Output:** $(\boldsymbol{x}, \boldsymbol{y})$ NE of rank-1 $(\mathbf{A}, \mathbf{B})$
1 **for** $k = 1, \ldots, K$ **do**
2    $(\boldsymbol{x}^{(k)}, \boldsymbol{y}^{(k)}) =$
    $\text{OMWU}(\boldsymbol{x}^{(k-1)}, \boldsymbol{y}^{(k-1)}, \lambda^{(k-1)})$
3    **if** $(\boldsymbol{x}^{(k)}, \boldsymbol{y}^{(k)})$ *is NE of* $(\mathbf{A}, \mathbf{B})$ **then**
4      **return** $(\boldsymbol{x}^{(k)}, \boldsymbol{y}^{(k)})$
5    **else**
6      $\lambda^{(k)} = \lambda^{(k-1)} - \frac{1}{2^k}\mathbf{sgn}\left(\boldsymbol{x}^{(k)^\top}\boldsymbol{a} - \lambda\right)$
     /* update $\lambda$ according to
     sign */
7

---

**Algorithm 2:** OMWU$(\boldsymbol{x}, \boldsymbol{y}, \lambda)$

**Input:** $(\boldsymbol{x}^{(0)}, \boldsymbol{y}^{(0)}) \in \Delta_n \times \Delta_m$, $\lambda \in [0, 1]$
**Output:** $(\boldsymbol{x}, \boldsymbol{y})$ such that $\text{Gap}(\boldsymbol{x}, \boldsymbol{y}) \leq \epsilon$
1 $\boldsymbol{g}_X^{(0)} = \mathbf{0}$
2 $\boldsymbol{g}_Y^{(0)} = \mathbf{0}$
3 **for** $t = 1, 2, \ldots, T$ **do**
4    $\boldsymbol{g}_X^{(t)} = -\nabla_{\boldsymbol{x}} f_\lambda(\boldsymbol{x}^{(t)}, \boldsymbol{y}^{(t)})$
5    $\boldsymbol{g}_Y^{(t)} = -\nabla_{\boldsymbol{y}} f_\lambda(\boldsymbol{x}^{(t)}, \boldsymbol{y}^{(t)})$
6    $\boldsymbol{x}_i^{(t+1)} = \dfrac{\boldsymbol{x}_i^{(t)}\exp(-\eta_X 2\boldsymbol{g}_{X,i}^{(t)} - \eta_X \boldsymbol{g}_{X,i}^{(t-1)})}{\sum_{i=1}^{n}\boldsymbol{x}_i^{(t)}\exp(-\eta_X 2\boldsymbol{g}_{X,i}^{(t)} - \eta_X \boldsymbol{g}_{X,i}^{(t-1)})}$

   $\boldsymbol{y}_j^{(t+1)} = \dfrac{\boldsymbol{y}_j^{(t)}\exp(-\eta_Y 2\boldsymbol{g}_{Y,j}^{(t)} - \eta_Y \boldsymbol{g}_{Y,j}^{(t-1)})}{\sum_{j=1}^{m}\boldsymbol{y}_j^{(t)}\exp(-\eta_Y 2\boldsymbol{g}_{Y,j}^{(t)} - \eta_X \boldsymbol{g}_{Y,j}^{(t-1)})}$

7 **return** $(\overline{\boldsymbol{x}}^{(T)}, \overline{\boldsymbol{y}}^{(T)})$

---

**Example 3.2** (Full Version in Example A.2). Let $\boldsymbol{z}^\star = (\boldsymbol{x}^\star, \boldsymbol{y}^\star)$ be a NE of the bimatrix rank-1 game $(\mathbf{A}, \mathbf{B}) = (\mathbf{A}, -\mathbf{A} + \boldsymbol{a}\boldsymbol{b}^\top)$. We show that there exists a game such that the Minty variational inequality (MVI) is not satisfied.

$$\text{(MVI)} \qquad \langle F(\boldsymbol{z}), \boldsymbol{z} - \boldsymbol{z}^\star \rangle \geq 0 \; \forall \boldsymbol{z} = \begin{bmatrix} \boldsymbol{x} \\ \boldsymbol{y} \end{bmatrix} \text{ where } F(\boldsymbol{z}) = \begin{bmatrix} -\mathbf{A}\boldsymbol{y} \\ -\mathbf{B}^\top \boldsymbol{x} \end{bmatrix}$$

We examine solutions $\boldsymbol{z}^\star = (\boldsymbol{x}^\star, \boldsymbol{y}^\star)$ that correspond to the Nash equilibrium of the selected game.

$$\langle F(\boldsymbol{z}), \boldsymbol{z} - \boldsymbol{z}^\star \rangle = \begin{bmatrix} -\mathbf{A}\boldsymbol{y}, & -\mathbf{B}^\top \boldsymbol{x} \end{bmatrix} \begin{bmatrix} \boldsymbol{x} - \boldsymbol{x}^\star \\ \boldsymbol{y} - \boldsymbol{y}^\star \end{bmatrix} = -\langle \boldsymbol{x} - \boldsymbol{x}^\star, \mathbf{A}\boldsymbol{y} \rangle - \langle \boldsymbol{y} - \boldsymbol{y}^\star, \mathbf{B}^\top \boldsymbol{x} \rangle$$

Let $\mathbf{A} = \begin{bmatrix} 1 & 0 \\ 0 & 1 \end{bmatrix}$ and $\mathbf{B} = \begin{bmatrix} 1 & -2 \\ -1 & 0 \end{bmatrix}$. This game has the two pure equilibria $(\boldsymbol{x}_1, \boldsymbol{y}_1) = ((1,0), (1,0))$ and $(\boldsymbol{x}_2, \boldsymbol{y}_2) = ((0,1), (0,1))$, and the mixed equilibrium $(\boldsymbol{x}_3, \boldsymbol{y}_3) = ((\frac{1}{4}, \frac{3}{4}), (\frac{1}{2}, \frac{1}{2}))$. However, none of these equilibria satisfy the Minty criterion.

### 3.2 MAIN RESULTS

In this section, we introduce our method, and sketch the main ingredients required for the proof of Theorem 3.7. To facilitate the presentation, we break it down into three main steps. The first step consists of showing that only specific *approximate* Nash equilibria of the parameterized zero-sum correspond to *approximate* Nash equilibria of the underlying rank-1 game. Then, we formulate a suitable min-max optimization problem whose stationary points are extendable –under conditions– to NE. Finally, we present an efficient algorithm for solving the optimization problem.

The following lemma relies on the correspondence established in Corollary 2.8. Although this appears to be applicable solely to *exact* Nash equilibria, we demonstrate that by relaxing both conditions, it also extends to *approximate* Nash equilibria.

**Lemma 3.3.** *Let $(\boldsymbol{x}, \boldsymbol{y})$ be an $\epsilon$-approximate Nash equilibrium of the zero sum game with payoff $(\mathbf{A} - \mathbb{1}\lambda\boldsymbol{b}^\top)$ and $|\boldsymbol{x}^\top \boldsymbol{a} - \lambda| \leq \epsilon$. Then $(\boldsymbol{x}, \boldsymbol{y})$ is an $(3\epsilon)$-approximate Nash equilibrium of the rank-1 game $(\mathbf{A}, -\mathbf{A} + \mathbb{1}\lambda\boldsymbol{b}^\top)$.*

A study of Lemma 3.3leads us to the following observation: to compute an approximate Nash equilibrium of the rank-1 game, we should seek a NE of the parameterized zero-sum game which simultaneously minimizes the distance $(\boldsymbol{x}^\top \boldsymbol{a} - \lambda)$. By virtue of this observation, we formulate the following min-max optimization problem; its full proof is deferred to Appendix A.

**Lemma 3.4.** *Consider the function $f_\lambda(\boldsymbol{x}, \boldsymbol{y}) \coloneqq \boldsymbol{x}^\top \left(\mathbf{A} - \mathbb{1}\lambda\boldsymbol{b}^\top\right)\boldsymbol{y} - \frac{1}{2}(\boldsymbol{x}^\top \boldsymbol{a} - \lambda)^2$. $f$ is concave with respect to $\boldsymbol{x}$ and convex (linear) with respect to $\boldsymbol{y}$. Moreover, assume that the corresponding zero-sum game has a $\epsilon$-approximate Nash equilibrium $(\boldsymbol{x}^\star, \boldsymbol{y}^\star)$ with $|\boldsymbol{x}^{*\top}\boldsymbol{a} - \lambda| \leq \epsilon$. Then, any $\epsilon$-approximate stationary point $(\tilde{\boldsymbol{x}}, \tilde{\boldsymbol{y}})$ of $f$ must satisfy the following.*

1. *$|\tilde{\boldsymbol{x}}^\top \boldsymbol{a} - \lambda| \leq \sqrt{5\epsilon}$.*

2. *$(\tilde{\boldsymbol{x}}, \tilde{\boldsymbol{y}})$ must be an $(6\sqrt{\epsilon})$-NE of the zero-sum game $(\mathbf{A} - \mathbb{1}\lambda\boldsymbol{b}^\top, -\mathbf{A} + \mathbb{1}\lambda\boldsymbol{b}^\top)$.*

*Sketch Proof.* Given that $(\tilde{\boldsymbol{x}}, \tilde{\boldsymbol{y}})$ is an $\epsilon$-approximate stationary point of $f$, it satisfies the following variational inequalities (VI).

VI (1) $\langle \nabla_{\boldsymbol{x}} f_\lambda(\tilde{\boldsymbol{x}}, \tilde{\boldsymbol{y}}), \boldsymbol{x}' - \tilde{\boldsymbol{x}} \rangle = \langle \left(\mathbf{A} - \mathbb{1}\lambda\boldsymbol{b}^\top\right)\tilde{\boldsymbol{y}} - \left(\boldsymbol{x}^\top\boldsymbol{a} - \lambda\right)\boldsymbol{a}, \boldsymbol{x}' - \tilde{\boldsymbol{x}} \rangle \leq \epsilon \ \forall \boldsymbol{x}' \in \Delta_n$

VI (2) $\langle \nabla_{\boldsymbol{y}} f_\lambda(\tilde{\boldsymbol{x}}, \tilde{\boldsymbol{y}}), \boldsymbol{y}' - \tilde{\boldsymbol{y}} \rangle = \langle \left(\mathbf{A} - \mathbb{1}\lambda\boldsymbol{b}^\top\right)^\top \tilde{\boldsymbol{x}}, \boldsymbol{y}' - \tilde{\boldsymbol{y}} \rangle \geq -\epsilon \quad \forall \boldsymbol{y}' \in \Delta_m$

First, we use the fact that $f(\cdot, \boldsymbol{y})$ is a concave function for any $\boldsymbol{y}$, which implies that it satisfies the inequality $f_\lambda(\boldsymbol{x}', \tilde{\boldsymbol{y}}) \leq f_\lambda(\tilde{\boldsymbol{x}}, \tilde{\boldsymbol{y}}) + \langle \nabla_{\boldsymbol{x}} f_\lambda(\tilde{\boldsymbol{x}}, \tilde{\boldsymbol{y}}), \boldsymbol{x}' - \tilde{\boldsymbol{x}} \rangle$. Therefore, when setting $\boldsymbol{x}' = \boldsymbol{x}^\star$ and from VI (1) and the fact $|\boldsymbol{x}^{\star\top}\boldsymbol{a} - \lambda| \leq \epsilon$ we obtain the following.

$$\boldsymbol{x}^{\star\top}\left(\mathbf{A} - \mathbb{1}\lambda\boldsymbol{b}^\top\right)\tilde{\boldsymbol{y}} \leq \tilde{\boldsymbol{x}}^\top\left(\mathbf{A} - \mathbb{1}\lambda\boldsymbol{b}^\top\right)\tilde{\boldsymbol{y}} - \frac{1}{2}\left(\tilde{\boldsymbol{x}}^\top\boldsymbol{a} - \lambda\right)^2 + 2\epsilon$$

Furthermore, $f_\lambda(\boldsymbol{x}, \cdot)$ is linear,for any $\boldsymbol{x}$, which implies that it satisfies the equality $f_\lambda(\tilde{\boldsymbol{x}}, \boldsymbol{y}') = f_\lambda(\tilde{\boldsymbol{x}}, \tilde{\boldsymbol{y}}) + \langle \nabla_{\boldsymbol{y}} f_\lambda(\tilde{\boldsymbol{x}}, \tilde{\boldsymbol{y}}), \boldsymbol{y}' - \tilde{\boldsymbol{y}} \rangle$ for any $\boldsymbol{y}' \in \Delta_m$. Therefore, when setting $\boldsymbol{y}' = \boldsymbol{y}^\star$ and from VI (2) we obtain the following.

$$\tilde{\boldsymbol{x}}^\top\left(\mathbf{A} - \mathbb{1}\lambda\boldsymbol{b}^\top\right)\boldsymbol{y}^\star \geq \tilde{\boldsymbol{x}}^\top\left(\mathbf{A} - \mathbb{1}\lambda\boldsymbol{b}^\top\right)\tilde{\boldsymbol{y}} - \epsilon$$

The last two inequalities follow from the approximate NE definition of $(\boldsymbol{x}^\star, \boldsymbol{y}^\star)$, with $\boldsymbol{x}$ representing the maximizer and $\boldsymbol{y}$ representing the minimizer.

$$\boldsymbol{x}^{\star\top}\left(\mathbf{A} - \mathbb{1}\lambda\boldsymbol{b}^\top\right)\boldsymbol{y}^\star \geq \tilde{\boldsymbol{x}}^\top\left(\mathbf{A} - \mathbb{1}\lambda\boldsymbol{b}^\top\right)\boldsymbol{y}^\star - \epsilon$$
$$\boldsymbol{x}^{\star\top}\left(\mathbf{A} - \mathbb{1}\lambda\boldsymbol{b}^\top\right)\boldsymbol{y}^\star \leq \boldsymbol{x}^{\star\top}\left(\mathbf{A} - \mathbb{1}\lambda\boldsymbol{b}^\top\right)\tilde{\boldsymbol{y}} + \epsilon$$

By combining the above four inequalities, we prove the first claim: $|\tilde{\boldsymbol{x}}^\top \boldsymbol{a} - \lambda| \leq \sqrt{5\epsilon}$. The second claim–$(\tilde{\boldsymbol{x}}, \tilde{\boldsymbol{y}})$ is a Nash Equilibrium (NE) of the parameterized zero-sum–follows from the VI (1), VI (2) alongside the inequalities we showed. $\square$

So far, we have verified that *any* stationary point $(\tilde{\boldsymbol{x}}, \tilde{\boldsymbol{y}})$ of $f_\lambda$ constitutes a Nash equilibrium for the rank-1 game under the condition that a NE does exists *for that particular $\lambda$*. In other words, Lemma 3.4 holds true only in the case where the parameter attains the correct value $\lambda^\star$–i.e., corresponding to a Nash equilibrium $(\boldsymbol{x}^\star, \boldsymbol{y}^\star)$ of the rank-1 game where $(\boldsymbol{x}^\star)^\top \boldsymbol{a} = \lambda^\star$. This raises the question of how we can determine the correct value of $\lambda$. Hopefully, the following simple but important lemma answers this question.

**Lemma 3.5** (Approximate Nash equilibria maintain sign)**.** *Let $(\tilde{\boldsymbol{x}}, \tilde{\boldsymbol{y}})$ and $(\boldsymbol{x}^\star, \boldsymbol{y}^\star)$ be $\epsilon$-approximate Nash equilibria of the parameterized zero-sum game $(\mathbf{A} - \mathbb{1}\lambda\boldsymbol{b}^\top, -\mathbf{A} + \mathbb{1}\lambda\boldsymbol{b}^\top)$, so that $\tilde{\boldsymbol{x}}^\top \boldsymbol{a} > \lambda$ and $\boldsymbol{x}^{\star\top}\boldsymbol{a} < \lambda$. Then, there exists $(\overline{\boldsymbol{x}}, \overline{\boldsymbol{y}})$ that is $\epsilon$-approximate NE with $\overline{\boldsymbol{x}}^\top \boldsymbol{a} = \lambda$.*

The lemma above answers our question in the following way: whenever a stationary point $(\boldsymbol{x}^\star, \boldsymbol{y}^\star)$ of $f_\lambda(\cdot, \cdot)$ fails to be an $\epsilon$-approximate Nash equilibria for the zero-sum and $|\boldsymbol{x}^\top \boldsymbol{a} - \lambda|$ be less than or equal to $\epsilon$, this necessarily means that there does not exist any NE that satisfies that equation *for that particular* $\lambda$; otherwise, any stationary point of $f_\lambda$ would suffice. More importantly, it implies that $\boldsymbol{x}^\top \boldsymbol{a} - \lambda$ must maintain its sign for any $\epsilon$-approximate Nash equilibrium in the zero-sum game for that particular $\lambda$. In other words, if there were a pair of Nash equilibria where $(\boldsymbol{x}^\top \boldsymbol{a} - \lambda)$ has opposite signs, then according to Lemma 3.5, it would guarantee the existence of a third solution where this difference equals zero.

In a similar spirit to Lemma 2.10, Lemma 3.5 serves as an invariant in our approach. Algorithm 1 performs binary search to parameter $\lambda$. At each iteration, it employs Algorithm 2 to obtain an $\epsilon$-approximate stationary point of $f_\lambda$. This solution either constitutes a Nash equilibrium of the rank-1 game $(\mathbf{A}, \mathbf{B})$ or indicates that no Nash equilibrium exists for that *particular* $\lambda$ and the *specified precision* $\epsilon$. In the latter case, the sign of the difference $(\boldsymbol{x}^\top \boldsymbol{a} - \lambda)$ determines the next value of $\lambda$. For instance, if $\boldsymbol{x}^\top \boldsymbol{a} - \lambda < -\epsilon$, then $\lambda$ is updated as $\lambda = \lambda - \frac{1}{2^k}$; otherwise, if $\boldsymbol{x}^\top \boldsymbol{a} - \lambda > +\epsilon$, then $\lambda$ is set as $\lambda + \frac{1}{2^k}$.

In the final part, we introduce an efficient method for obtaining an $\epsilon$-approximate stationary point of $f_\lambda$ for a specific $\lambda$. Leveraging the structure of $f_\lambda$, we utilize an Optimistic Multiplicative Weights Update (OMWU), as detailed in Daskalakis & Panageas (2018a). This method offers two key advantages: it does not require a projection step, which can be computationally expensive, and it achieves faster convergence rates by incorporating optimism. A more comprehensive presentation of optimistic methods is deferred to Appendix B. Lemma 3.6 provides us with a specific number of iterations required.

**Lemma 3.6.** *Let* $(\mathbf{x}^{(0)}, \mathbf{y}^{(0)})$ *be initialized as uniform vectors* $(\mathbb{1}/n, \mathbb{1}/m)$. *Suppose that both players employ OMWU for* $T = \Omega\left(\frac{(\ln n + \ln m)}{\epsilon}\right)$ *with learning rates* $\eta_X = \eta_Y = \frac{1}{16\sqrt{2}\max\{\|\mathbf{A}\|_2, \|\boldsymbol{a}\|_2\}}$ *and fixed* $\lambda$. *Then, Algorithm 2 returns a pair of strategies* $\left(\overline{\boldsymbol{x}}^{(T)}, \overline{\boldsymbol{y}}^{(T)}\right)$ *that guarantees the following.*

$$Gap\left(\overline{\boldsymbol{x}}^{(T)}, \overline{\boldsymbol{y}}^{(T)}\right) = \max_{\boldsymbol{x} \in \Delta_n} f_\lambda\left(\boldsymbol{x}, \overline{\boldsymbol{y}}^{(T)}\right) - \min_{\boldsymbol{y} \in \Delta_m} f_\lambda\left(\overline{\boldsymbol{x}}^{(T)}, \boldsymbol{y}\right) \leq \epsilon.$$

*Also, it holds that* $\left(\overline{\boldsymbol{x}}^{(T)}, \overline{\boldsymbol{y}}^{(T)}\right)$ *is an $\epsilon$-approximate stationary point of* $f_\lambda$.

We are now prepared to present the main technical theorem that integrates all the components established thus far into a concise statement

**Theorem 3.7** (Main Result). *Let* $(\mathbf{A}, \mathbf{B})$ *be a rank-1 game. Suppose that both players employ OMWU for* $T = \Omega\left(\frac{(\log n + \log m)}{\epsilon^2} \log(\frac{1}{\epsilon})\right)$. *Then, Algorithm 1 returns a pair of strategies* $(\boldsymbol{x}^\star, \boldsymbol{y}^\star)$ *that constitutes an $18\epsilon$-approximate Nash equilibrium of the rank-1 game.*

## 4 CONCLUSION

In this work, we examined the structural properties, as well as challenges, of rank-1 games, an important class the generalizes over the well-understood zero-sum games. We introduced a decentralized learning procedure that provable converges to a Nash equilibrium of the game. Nevertheless, a number of important questions remain open. Proving a matching lower bound will certify the optimality of our solution. However, given that our approach hinges on the idea of reducing a rank-1 game into a zero-sum game and then optimizing a carefully crafted minmax problem, we presume that, unless an entirely different methodology or accelerated methods are considered, our solution remains optimal. Furthermore, a fundamental question is whether our approach, or one similar to it, can be extended further for rank-2 games.

## 5 ACKNOWLEDGEMENTS

We are grateful to anonymous reviewers at ICLR for valuable feedback. Ioannis Panageas would like to acknowledge startup grant from UCI and UCI ICS Research Award. Part of this work was conducted while Nikolas and Ioannis were visiting Archimedes Research Unit. This work has been partially supported by project MIS 5154714 of the National Recovery and Resilience Plan Greece 2.0 funded by the European Union under the NextGenerationEU Program.

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

## A APPENDIX

### A.1 TWO-EXAMPLES OF SECTION 3.1

**Example A.1.** In this example, we demonstrate that an approximate Nash equilibrium of the zero-sum game $(\mathbf{A} - \mathbb{1}\lambda\boldsymbol{b}^\top, -\mathbf{A} + \mathbb{1}\lambda\boldsymbol{b}^\top)$ does not constitute an approximate Nash Equilibrium of the rank-1 game, even when we know a *correct* value of $\lambda$.

Let $\mathbf{A} - \mathbb{1}\lambda\boldsymbol{b}^\top = \begin{bmatrix} 1 & 1 \\ 1-\epsilon & 1-\epsilon \end{bmatrix}$. This game has an exact NE of the form $(\boldsymbol{x}^\star, \boldsymbol{y}^\star) = ((1,0), (\mu, 1-\mu))$ for any $\mu \in [0,1]$. Additionally, $(\tilde{\boldsymbol{x}}, \tilde{\boldsymbol{y}}) = ((\frac{1}{2}, \frac{1}{2}), (\frac{1}{2}, \frac{1}{2}))$ is an $\frac{\epsilon}{2}$-approximate Nash equilibrium of the zero-sum game $(\mathbf{A} - \mathbb{1}\lambda\boldsymbol{b}^\top, -\mathbf{A} + \mathbb{1}\lambda\boldsymbol{b}^\top)$

$$\boldsymbol{x}^{\star\top} \left(\mathbf{A} - \mathbb{1}\lambda\boldsymbol{b}^\top\right) \boldsymbol{y}^\star = 1 \text{ and } \tilde{\boldsymbol{x}}^\top \left(\mathbf{A} - \mathbb{1}\lambda\boldsymbol{b}^\top\right) \tilde{\boldsymbol{y}} = \begin{bmatrix} \frac{1}{2} & \frac{1}{2} \end{bmatrix} \begin{bmatrix} 1 & 1 \\ 1-\epsilon & 1-\epsilon \end{bmatrix} \begin{bmatrix} \frac{1}{2} \\ \frac{1}{2} \end{bmatrix} = 1 - \frac{\epsilon}{2}$$

We now choose $\boldsymbol{a} = \begin{bmatrix} 1 & 2 \end{bmatrix}^\top$ and $\boldsymbol{b} = \begin{bmatrix} 2 & 1 \end{bmatrix}^\top$. Thus, setting $\lambda = 1 \left(:= \boldsymbol{x}^{\star\top}\boldsymbol{a}\right)$ the rank-1 game is defined as follows.

$$\mathbf{A} = \begin{bmatrix} 1 & 1 \\ 1-\epsilon & 1-\epsilon \end{bmatrix} + \begin{bmatrix} 2 & 1 \\ 2 & 1 \end{bmatrix} = \begin{bmatrix} 3 & 2 \\ 3-\epsilon & 2-\epsilon \end{bmatrix} \quad \text{and} \quad \mathbf{B} = \begin{bmatrix} -1 & -1 \\ 1+\epsilon & \epsilon \end{bmatrix}$$

We confirm that $(\tilde{\boldsymbol{x}}, \tilde{\boldsymbol{y}}) = \left(\left(\frac{1}{2}, \frac{1}{2}\right), \left(\frac{1}{2}, \frac{1}{2}\right)\right)$ no longer qualifies as an $\frac{\epsilon}{2}$-approximate Nash equilibrium for the original rank-1 game as the condition for the column player does not hold.

$$\tilde{\boldsymbol{x}}^\top \mathbf{B}\tilde{\boldsymbol{y}} - \tilde{\boldsymbol{x}}^\top \mathbf{B} \begin{bmatrix} 1 \\ 0 \end{bmatrix} = \left(-\frac{1}{4} + \frac{\epsilon}{2}\right) - \left(\frac{\epsilon}{2}\right) = -\frac{1}{4}$$

Specifically, this example highlights that irrespective of our knowledge of $\lambda$, an approximate NE of the zero-sum game that does not satisfy $\boldsymbol{x}^\top \boldsymbol{a} \approx \lambda$, cannot be a Nash Equilibrium in the original rank-1 game.

**Example A.2.** Let $\boldsymbol{z}^\star = (\boldsymbol{x}^\star, \boldsymbol{y}^\star)$ be a NE of the bimatrix rank-1 game $(\mathbf{A}, \mathbf{B}) = (\mathbf{A}, -\mathbf{A} + \boldsymbol{a}\boldsymbol{b}^\top)$. We introduce the operator $F$,

$$F(\boldsymbol{z}) = F\left(\begin{bmatrix} \boldsymbol{x} \\ \boldsymbol{y} \end{bmatrix}\right) = F\begin{bmatrix} -\mathbf{A}\boldsymbol{y} \\ -\mathbf{B}^\top\boldsymbol{x} \end{bmatrix},$$

i.e. the gradient operator with respect to the underlying rank 1 game. We note that as both players aim to maximize their payoff, the considered gradients come with a negative sign.

As described in Section 2.3, a recent literature line investigates structural properties of games that enable learning procedure to converge to a Nash equilibrium, beyond the case of a monotone operator $F$. An example of compelling criterion that guarantees convergence is the existence of solution to the Minty variational inequality (MVI) problem Mertikopoulos et al. (2018) or the weak MVI problem Diakonikolas et al. (2021). In this example, we show that there exists a game such that neither of those criterion does not hold.

$$\text{(MVI)} \quad \langle F(\boldsymbol{z}), \boldsymbol{z} - \boldsymbol{z}^\star \rangle \geq 0 \text{ for all } \boldsymbol{z} = \begin{bmatrix} \boldsymbol{x} \\ \boldsymbol{y} \end{bmatrix}$$

$$\text{(weak MVI)} \quad \langle F(\boldsymbol{z}), \boldsymbol{z} - \boldsymbol{z}^\star \rangle \geq -\frac{\rho}{2}\|F(\boldsymbol{z})\|_2^2 \text{ for all } \boldsymbol{z} = \begin{bmatrix} \boldsymbol{x} \\ \boldsymbol{y} \end{bmatrix} \text{ where } \rho \in \left[0, \frac{1}{4L}\right)$$

The parameter $L$ is the Lipschitz constant of the operator $F$ with respect to the $\ell_2$ norm: $\|F(z) - F(z')\|_2 \leq L\|z - z'\|_2$. Now, as the set of solutions to the (weak) MVI is a subset to the solutions set of (SVI), we only have to examine $z^\star = (x^\star, y^\star)$ that correspond to the Nash equilibrium of the following game.

$$\mathbf{A} = \begin{bmatrix} 1 & 0 \\ 0 & 1 \end{bmatrix} \quad \mathbf{B} = \begin{bmatrix} 1 & -2 \\ -1 & 0 \end{bmatrix}$$

$$\langle F(z), z - z^\star \rangle = \begin{bmatrix} -\mathbf{A}y, & -\mathbf{B}^\top x \end{bmatrix} \begin{bmatrix} x - x^\star \\ y - y^\star \end{bmatrix} = -\langle x - x^\star, \mathbf{A}y \rangle - \langle y - y^\star, \mathbf{B}^\top x \rangle$$

To calculate the Lipschitz constant, we can simply find the spectral norm of the block matrix $C = \begin{bmatrix} \mathbf{0} & -A \\ -B^\top & \mathbf{0} \end{bmatrix}$; that is, $L = \|C\|_2 = 2.288$. This game has the two pure equilibria $(x_1, y_1) = ((1,0),(1,0))$ and $(x_2, y_2) = ((0,1),(0,1))$, and the mixed equilibrium $(x_3, y_3) = ((\frac{1}{4}, \frac{3}{4}),(\frac{1}{2}, \frac{1}{2}))$. It is important to note that when setting $z = (x, y)$ to be another NE, the inequality must necessarily be less than or equal to zero ($\leq$), as the constituent terms (without the minus sign) are greater than or equal to zero ($\geq$) by the definition of NE.

- $(x^\star, y^\star) = (x_1, y_1)$ and $(x, y) = (x_2, y_2)$.

$$\langle x_2 - x_1, \mathbf{A}y_2 \rangle = \begin{bmatrix} -1 & 1 \end{bmatrix} \begin{bmatrix} 0 \\ 1 \end{bmatrix} = 1 > 0$$

- $(x^\star, y^\star) = (x_2, y_2)$ and $(x, y) = (x_1, y_1)$.

$$\langle x_1 - x_2, \mathbf{A}y_1 \rangle = \begin{bmatrix} 1 & -1 \end{bmatrix} \begin{bmatrix} 1 \\ 0 \end{bmatrix} = 1 > 0$$

- $(x^\star, y^\star) = (x_3, y_3)$ and $(x, y) = (x_1, y_1)$.

$$\langle x_1 - x_3, \mathbf{A}y_1 \rangle = \begin{bmatrix} \frac{3}{4} & -\frac{3}{4} \end{bmatrix} \begin{bmatrix} 1 \\ 0 \end{bmatrix} = \frac{3}{4} > 0$$

Repeating the same procedure for the second term $\langle y - y^\star, \mathbf{B}^\top x \rangle$, we get that $\langle F(z_2), z_2 - z_1 \rangle = -2$, $\langle F(z_1), z_1 - z_2 \rangle = -4$, $\langle F(z_1), z_1 - z_3 \rangle = -9/4$. Furthermore, it is relative straightforward to verify that $\|F(z_1)\|_2^2 = 6$, $\|F(z_2)\|_2^2 = 2$, $\|F(z_3)\|_2^2 = 1$. We readily conclude that neither the Minty or weak Minty criterion is satisfied.

## A.2 PROOF OF LEMMA 3.3

**Lemma A.3** (Proof of Lemma 3.3). *Let $(x, y)$ be an $\epsilon$-approximate Nash equilibrium of the zero sum game with payoff $(\mathbf{A} - \mathbb{1}\lambda b^\top)$ and $|x^\top a - \lambda| \leq \epsilon$. Then $(x, y)$ is an $(3\epsilon)$-approximate Nash equilibrium of the rank-1 game $(\mathbf{A}, -\mathbf{A} + \mathbb{1}\lambda b^\top)$.*

*Proof.* From the definition of the Nash equilibrium for the row player, it holds that

$$\boldsymbol{x}^\top \left(\mathbf{A} - \mathbb{1}\lambda\boldsymbol{b}^\top\right) \boldsymbol{y} \geq \boldsymbol{x}'^\top \left(\mathbf{A} - \mathbb{1}\lambda\boldsymbol{b}^\top\right) \boldsymbol{y} - \epsilon \quad \text{for every } \boldsymbol{x}' \in \Delta_n$$

Since any $\boldsymbol{x} \in \Delta_n$ satisfies $\boldsymbol{x}^\top \mathbb{1} = 1$, it follows that $\boldsymbol{x}^\top \mathbb{1}(\lambda\boldsymbol{b}^\top) = (\boldsymbol{x}^\top \mathbb{1})(\lambda\boldsymbol{b}^\top) = (\lambda\boldsymbol{b}^\top)$.

$$\boldsymbol{x}^\top \mathbf{A}\boldsymbol{y} \geq \boldsymbol{x}'^\top \mathbf{A}\boldsymbol{y} - \epsilon \quad \text{for every } \boldsymbol{x}' \in \Delta_n \tag{3}$$

Thus, Equation (3) satisfies the Nash equilibrium condition of row player for the rank-1 game $(\mathbf{A}, -\mathbf{A}+\boldsymbol{a}\boldsymbol{b}^\top)$; hence they play approximate best response. Similarly, the Nash equilibrium condition for the column player yields the following.

$$\boldsymbol{x}^\top \left(\mathbf{A} - \mathbb{1}\lambda\boldsymbol{b}^\top\right) \boldsymbol{y} \leq \boldsymbol{x}^\top \left(\mathbf{A} - \mathbb{1}\lambda\boldsymbol{b}^\top\right) \boldsymbol{y}' + \epsilon$$

$$\Rightarrow \qquad \boldsymbol{x}^\top \mathbf{A}\boldsymbol{y} \leq \boldsymbol{x}^\top \mathbf{A}\boldsymbol{y}' - \lambda\boldsymbol{b}^\top(\boldsymbol{y} - \boldsymbol{y}') + \epsilon$$

$$\Rightarrow \qquad \boldsymbol{x}^\top \mathbf{A}\boldsymbol{y} \leq \boldsymbol{x}^\top \mathbf{A}\boldsymbol{y}' + \left(-\boldsymbol{x}^\top \boldsymbol{a} + \epsilon\right)\boldsymbol{b}^\top(\boldsymbol{y} - \boldsymbol{y}') + \epsilon$$

$$\Rightarrow \qquad \boldsymbol{x}^\top \mathbf{A}\boldsymbol{y} \leq \boldsymbol{x}^\top \mathbf{A}\boldsymbol{y}' - \boldsymbol{x}^\top \boldsymbol{a}\boldsymbol{b}^\top(\boldsymbol{y} - \boldsymbol{y}') + \epsilon\boldsymbol{b}^\top(\boldsymbol{y} - \boldsymbol{y}') + \epsilon$$

$$\Rightarrow \qquad \boldsymbol{x}^\top \left(\mathbf{A} - \boldsymbol{a}\boldsymbol{b}^\top\right)\boldsymbol{y} \leq \boldsymbol{x}^\top \left(\mathbf{A} - \boldsymbol{a}\boldsymbol{b}^\top\right)\boldsymbol{y}' + 3\epsilon \tag{4}$$

The last inequality follows from application the Cauchy-Schwarz [3] inequality and the triangle inequality: $\langle \boldsymbol{b}, \boldsymbol{y} - \boldsymbol{y}'\rangle \leq \|\boldsymbol{b}\|_\infty \|\boldsymbol{y} - \boldsymbol{y}'\|_1 \leq \max_{j \in [m]} |\boldsymbol{b}_j| (\|\boldsymbol{y}\|_1 + \|\boldsymbol{y}'\|_1) = 2$. Equations (3) and (4) prove that $(\boldsymbol{x}, \boldsymbol{y})$ satisfy the NE conditions of the game $(\mathbf{A}, -\mathbf{A} + \boldsymbol{a}\boldsymbol{b}^\top)$. Thus, $(\boldsymbol{x}, \boldsymbol{y})$ is an $(3\epsilon)$-approximate NE. $\qquad\square$

### A.3 PROOF OF LEMMA 3.4

**Lemma A.4** (Proof of Lemma 3.4). *Consider the function* $f_\lambda(\boldsymbol{x}, \boldsymbol{y}) := \boldsymbol{x}^\top \left(\mathbf{A} - \mathbb{1}\lambda\boldsymbol{b}^\top\right)\boldsymbol{y} - \frac{1}{2}(\boldsymbol{x}^\top \boldsymbol{a} - \lambda)^2$. *$f$ is concave with respect to $\boldsymbol{x}$ and convex (linear) with respect to $\boldsymbol{y}$. Moreover, assume that the corresponding zero-sum game has a $\epsilon$-approximate Nash equilibrium $(\boldsymbol{x}^\star, \boldsymbol{y}^\star)$ with $|\boldsymbol{x}^{*\top}\boldsymbol{a}-\lambda| \leq \epsilon$. Then, any $\epsilon$-approximate stationary point $(\tilde{\boldsymbol{x}}, \tilde{\boldsymbol{y}})$ of $f$ must satisfy the following.*

1. *$|\tilde{\boldsymbol{x}}^\top \boldsymbol{a} - \lambda| \leq \sqrt{5\epsilon}$.*

2. *$(\tilde{\boldsymbol{x}}, \tilde{\boldsymbol{y}})$ must be an $(6\sqrt{\epsilon})$-NE of the zero-sum game $(\mathbf{A} - \mathbb{1}\lambda\boldsymbol{b}^\top, -\mathbf{A} + \mathbb{1}\lambda\boldsymbol{b}^\top)$.*

*Proof.* Given that $(\tilde{\boldsymbol{x}}, \tilde{\boldsymbol{y}})$ is an $\epsilon$-approximate stationary point of $f$, it satisfies the following variational inequalities (VI).

VI (1) $\langle \nabla_{\boldsymbol{x}} f_\lambda(\tilde{\boldsymbol{x}}, \tilde{\boldsymbol{y}}), \boldsymbol{x}' - \tilde{\boldsymbol{x}}\rangle = \langle \left(\mathbf{A} - \mathbb{1}\lambda\boldsymbol{b}^\top\right)\tilde{\boldsymbol{y}} - \left(\boldsymbol{x}^\top \boldsymbol{a} - \lambda\right)\boldsymbol{a}, \boldsymbol{x}' - \tilde{\boldsymbol{x}}\rangle \leq \epsilon \ \forall \boldsymbol{x}' \in \Delta_n$

VI (2) $\langle \nabla_{\boldsymbol{y}} f_\lambda(\tilde{\boldsymbol{x}}, \tilde{\boldsymbol{y}}), \boldsymbol{y}' - \tilde{\boldsymbol{y}}\rangle = \langle \left(\mathbf{A} - \mathbb{1}\lambda\boldsymbol{b}^\top\right)^\top \tilde{\boldsymbol{x}}, \boldsymbol{y}' - \tilde{\boldsymbol{y}}\rangle \geq -\epsilon \quad \forall \boldsymbol{y}' \in \Delta_m$

First, we leverage the fact that $f(\cdot, \boldsymbol{y})$ is a concave function for any $\boldsymbol{y}$, which implies that it satisfies the inequality $f_\lambda(\boldsymbol{x}', \tilde{\boldsymbol{y}}) \leq f_\lambda(\tilde{\boldsymbol{x}}, \tilde{\boldsymbol{y}}) + \langle \nabla_{\boldsymbol{x}} f_\lambda(\tilde{\boldsymbol{x}}, \tilde{\boldsymbol{y}}), \boldsymbol{x}' - \tilde{\boldsymbol{x}}\rangle$. Therefore, when setting $\boldsymbol{x}' = \boldsymbol{x}^\star$ and from VI (1) and the fact $|\boldsymbol{x}^{\star\top}\boldsymbol{a} - \lambda| \leq \epsilon$ we obtain the following.

$$\boldsymbol{x}^{\star\top}\left(\mathbf{A} - \mathbb{1}\lambda\boldsymbol{b}^\top\right)\tilde{\boldsymbol{y}} - \frac{1}{2}\left(\boldsymbol{x}^{\star\top}\boldsymbol{a} - \lambda\right)^2 \leq \tilde{\boldsymbol{x}}^\top \left(\mathbf{A} - \mathbb{1}\lambda\boldsymbol{b}^\top\right)\tilde{\boldsymbol{y}} - \frac{1}{2}\left(\tilde{\boldsymbol{x}}^\top \boldsymbol{a} - \lambda\right)^2 + \epsilon$$

---

[3]In its general form, the Cauchy-Schwarz inequality states $|\langle \boldsymbol{a}, \boldsymbol{b}\rangle| \leq \|\boldsymbol{a}\|\|\boldsymbol{b}\|_*$, where $\|\cdot\|$ is any norm and $\|\cdot\|_*$ its dual.

$$\boldsymbol{x}^{\star\top}\left(\mathbf{A}-\mathbb{1}\lambda\boldsymbol{b}^\top\right)\tilde{\boldsymbol{y}}-\frac{\epsilon^2}{2}\le\tilde{\boldsymbol{x}}^\top\left(\mathbf{A}-\mathbb{1}\lambda\boldsymbol{b}^\top\right)\tilde{\boldsymbol{y}}-\frac{1}{2}\left(\tilde{\boldsymbol{x}}^\top\boldsymbol{a}-\lambda\right)^2+\epsilon$$

$$\boldsymbol{x}^{\star\top}\left(\mathbf{A}-\mathbb{1}\lambda\boldsymbol{b}^\top\right)\tilde{\boldsymbol{y}}\le\tilde{\boldsymbol{x}}^\top\left(\mathbf{A}-\mathbb{1}\lambda\boldsymbol{b}^\top\right)\tilde{\boldsymbol{y}}-\frac{1}{2}\left(\tilde{\boldsymbol{x}}^\top\boldsymbol{a}-\lambda\right)^2+2\epsilon \tag{5}$$

Furthermore, $f_\lambda(\boldsymbol{x},\cdot)$ is linear,for any $\boldsymbol{x}$, which implies that it satisfies the equality $f_\lambda(\tilde{\boldsymbol{x}},\boldsymbol{y}')=f_\lambda(\tilde{\boldsymbol{x}},\tilde{\boldsymbol{y}})+\langle\nabla_{\boldsymbol{y}}f_\lambda(\tilde{\boldsymbol{x}},\tilde{\boldsymbol{y}}),\boldsymbol{y}'-\tilde{\boldsymbol{y}}\rangle$ for any $\boldsymbol{y}'\in\Delta_m$. Therefore, when setting $\boldsymbol{y}'=\boldsymbol{y}^\star$ and from VI (2) we obtain the following.

$$\tilde{\boldsymbol{x}}^\top\left(\mathbf{A}-\mathbb{1}\lambda\boldsymbol{b}^\top\right)\boldsymbol{y}^\star-\frac{1}{2}\left(\tilde{\boldsymbol{x}}^\top\boldsymbol{a}-\lambda\right)^2\ge\tilde{\boldsymbol{x}}^\top\left(\mathbf{A}-\mathbb{1}\lambda\boldsymbol{b}^\top\right)\tilde{\boldsymbol{y}}-\frac{1}{2}\left(\tilde{\boldsymbol{x}}^\top\boldsymbol{a}-\lambda\right)^2-\epsilon$$

$$\Rightarrow\qquad \tilde{\boldsymbol{x}}^\top\left(\mathbf{A}-\mathbb{1}\lambda\boldsymbol{b}^\top\right)\boldsymbol{y}^\star\ge\tilde{\boldsymbol{x}}^\top\left(\mathbf{A}-\mathbb{1}\lambda\boldsymbol{b}^\top\right)\tilde{\boldsymbol{y}}-\epsilon \tag{6}$$

The last two inequalities follow from the approximate NE definition of $(\boldsymbol{x}^\star,\boldsymbol{y}^\star)$, with $\boldsymbol{x}$ representing the maximizer and $\boldsymbol{y}$ representing the minimizer.

$$\boldsymbol{x}^{\star\top}\left(\mathbf{A}-\mathbb{1}\lambda\boldsymbol{b}^\top\right)\boldsymbol{y}^\star\ge\tilde{\boldsymbol{x}}^\top\left(\mathbf{A}-\mathbb{1}\lambda\boldsymbol{b}^\top\right)\boldsymbol{y}^\star-\epsilon \tag{7}$$

$$\boldsymbol{x}^{\star\top}\left(\mathbf{A}-\mathbb{1}\lambda\boldsymbol{b}^\top\right)\boldsymbol{y}^\star\le\boldsymbol{x}^{\star\top}\left(\mathbf{A}-\mathbb{1}\lambda\boldsymbol{b}^\top\right)\tilde{\boldsymbol{y}}+\epsilon \tag{8}$$

We now have to combine appropriately the above four inequalities to prove the claim.

$$\Rightarrow (6)\qquad \tilde{\boldsymbol{x}}^\top\left(\mathbf{A}-\mathbb{1}\lambda\boldsymbol{b}^\top\right)\tilde{\boldsymbol{y}}\le\tilde{\boldsymbol{x}}^\top\left(\mathbf{A}-\mathbb{1}\lambda\boldsymbol{b}^\top\right)\boldsymbol{y}^\star+\epsilon$$

$$\Rightarrow (7)\qquad\qquad\qquad\le\boldsymbol{x}^{\star\top}\left(\mathbf{A}-\mathbb{1}\lambda\boldsymbol{b}^\top\right)\boldsymbol{y}^\star+2\epsilon$$

$$\Rightarrow (8)\qquad\qquad\qquad\le\boldsymbol{x}^{\star\top}\left(\mathbf{A}-\mathbb{1}\lambda\boldsymbol{b}^\top\right)\tilde{\boldsymbol{y}}+3\epsilon$$

$$\Rightarrow (5)\qquad\qquad\qquad\le\tilde{\boldsymbol{x}}^\top\left(\mathbf{A}-\mathbb{1}\lambda\boldsymbol{b}^\top\right)\tilde{\boldsymbol{y}}-\frac{1}{2}\left(\tilde{\boldsymbol{x}}^\top\boldsymbol{a}-\lambda\right)^2+5\epsilon$$

$$\Rightarrow\qquad \tilde{\boldsymbol{x}}^\top\left(\mathbf{A}-\mathbb{1}\lambda\boldsymbol{b}^\top\right)\tilde{\boldsymbol{y}}\le\tilde{\boldsymbol{x}}^\top\left(\mathbf{A}-\mathbb{1}\lambda\boldsymbol{b}^\top\right)\tilde{\boldsymbol{y}}-\frac{1}{2}\left(\tilde{\boldsymbol{x}}^\top\boldsymbol{a}-\lambda\right)^2+5\epsilon$$

$$\Rightarrow\qquad |\tilde{\boldsymbol{x}}^\top\boldsymbol{a}-\lambda|\le\sqrt{5\epsilon}$$

The fact that $(\tilde{\boldsymbol{x}},\tilde{\boldsymbol{y}})$ constitutes a Nash Equilibrium (NE) of the parameterized zero-sum arises from the VI (1), VI (2) in conjunction with the inequality we have just established.

$$\text{VI (2)}\qquad \langle\left(\mathbf{A}-\mathbb{1}\lambda\boldsymbol{b}^\top\right)^\top\tilde{\boldsymbol{x}},\boldsymbol{y}'-\tilde{\boldsymbol{y}}\rangle\ge-\epsilon\Rightarrow\tilde{\boldsymbol{x}}\left(\mathbf{A}-\mathbb{1}\lambda\boldsymbol{b}^\top\right)\tilde{\boldsymbol{y}}\le\tilde{\boldsymbol{x}}\left(\mathbf{A}-\mathbb{1}\lambda\boldsymbol{b}^\top\right)\boldsymbol{y}'+\epsilon$$

$$\text{VI (1)}\qquad \langle\left(\mathbf{A}-\mathbb{1}\lambda\boldsymbol{b}^\top\right)\tilde{\boldsymbol{y}}-\left(\tilde{\boldsymbol{x}}^\top\boldsymbol{a}-\lambda\right)\boldsymbol{a},\boldsymbol{x}'-\tilde{\boldsymbol{x}}\rangle\le\epsilon$$

$$\Rightarrow\qquad \tilde{\boldsymbol{x}}^\top\left(\mathbf{A}-\mathbb{1}\lambda\boldsymbol{b}^\top\right)\tilde{\boldsymbol{y}}\ge\boldsymbol{x}'^\top\left(\mathbf{A}-\mathbb{1}\lambda\boldsymbol{b}^\top\right)\tilde{\boldsymbol{y}}-(\tilde{\boldsymbol{x}}^\top\boldsymbol{a}-\lambda)\langle\boldsymbol{a},\boldsymbol{x}'-\tilde{\boldsymbol{x}}\rangle-\epsilon$$

$$\Rightarrow\qquad \tilde{\boldsymbol{x}}^\top\left(\mathbf{A}-\mathbb{1}\lambda\boldsymbol{b}^\top\right)\tilde{\boldsymbol{y}}\ge\boldsymbol{x}'^\top\left(\mathbf{A}-\mathbb{1}\lambda\boldsymbol{b}^\top\right)\tilde{\boldsymbol{y}}-2\sqrt{5\epsilon}-\epsilon \tag{9}$$

$$\Rightarrow\qquad \tilde{\boldsymbol{x}}^\top\left(\mathbf{A}-\mathbb{1}\lambda\boldsymbol{b}^\top\right)\tilde{\boldsymbol{y}}\ge\boldsymbol{x}'^\top\left(\mathbf{A}-\mathbb{1}\lambda\boldsymbol{b}^\top\right)\tilde{\boldsymbol{y}}-6\sqrt{\epsilon}$$

Equation (9) follows again from the Cauchy-Schwarz inequality along with the fact that $|\boldsymbol{x}^\top\boldsymbol{a}-\lambda|\le\sqrt{5\epsilon}$. Therefore, we conclude that $(\tilde{\boldsymbol{x}},\tilde{\boldsymbol{y}})$ is an $(6\sqrt{\epsilon})$-approximate NE of the zero-sum game $(\mathbf{A}-\mathbb{1}\lambda\boldsymbol{b}^\top,-\mathbf{A}+\mathbb{1}\lambda\boldsymbol{b}^\top)$. $\qquad\square$

### A.4 PROOF OF LEMMA 3.5

**Lemma A.5** (Proof of Lemma 3.5). *Let $(\tilde{\boldsymbol{x}}, \tilde{\boldsymbol{y}})$ and $(\boldsymbol{x}^\star, \boldsymbol{y}^\star)$ be $\epsilon$-approximate Nash equilibria of the parameterized zero-sum game $(\mathbf{A} - \mathbb{1}\lambda\boldsymbol{b}^\top, -\mathbf{A} + \mathbb{1}\lambda\boldsymbol{b}^\top)$, so that $\tilde{\boldsymbol{x}}^\top\boldsymbol{a} > \lambda$ and $\boldsymbol{x}^{\star\top}\boldsymbol{a} < \lambda$. Then, there exists $(\overline{\boldsymbol{x}}, \overline{\boldsymbol{y}})$ that is $\epsilon$-approximate NE with $\overline{\boldsymbol{x}}^\top\boldsymbol{a} = \lambda$.*

*Proof.* It is a well-established fact that the set of (approximate) Nash equilibria (NE) in a zero-sum game forms a convex set. In other words, any convex combination $\mu(\tilde{\boldsymbol{x}}, \tilde{\boldsymbol{y}}) + (1-\mu)(\boldsymbol{x}^\star, \boldsymbol{y}^\star)$, where $\mu \in [0,1]$, also constitutes a NE. Consequently, we consider a combination denoted as $\overline{\boldsymbol{x}}$, with the property that $\overline{\boldsymbol{x}}^\top\mathbf{a} = \lambda$. As explained earlier, this corresponding convex combination qualifies as an $\epsilon$-NE. □

### A.5 PROOF OF MAIN RESULT

**Theorem A.6** (Proof of Theorem 3.7). *Let $(\mathbf{A}, \mathbf{B})$ be a rank-1 game. Suppose that both players employ OMWU for $T = \Omega\left(\frac{(\log n + \log m)}{\epsilon^2}\log(\frac{1}{\epsilon})\right)$. Then, Algorithm 1 returns a pair of strategies $(\boldsymbol{x}^\star, \boldsymbol{y}^\star)$ that constitutes an $18\epsilon$-approximate Nash equilibrium of the rank-1 game.*

*Proof.* For a specific $\lambda$, we consider two cases. First, assuming that $\lambda$ is correct, meaning that there exists an $O(\epsilon)$-approximate Nash equilibrium for the rank-1 game, if we run Algorithm 2 for $T \geq O\left(\frac{(\log n + \log m)}{\epsilon^2}\right)$ iterations, we obtain a pair $(\tilde{\boldsymbol{x}}, \tilde{\boldsymbol{y}})$ that is an $\epsilon^2$-approximate point of $f_\lambda$. By combining Lemma 3.4 and Lemma 3.3, this pair constitutes a $18\epsilon$-approximate Nash equilibrium of the rank-1 game.

Conversely, if the value of $\lambda$ is incorrect, it must be updated. Considering the sign of the difference $(\boldsymbol{x}^\top\boldsymbol{a} - \lambda)$, the parameter is updated according to the rule in Algorithm 1. Since Algorithm 1 needs at most $K = \log\left(\frac{1}{\epsilon}\right)$ binary search steps, we conclude the aforementioned complexity. □

## B MIN-MAX OPTIMIZATION

### B.1 BASIC BACKGROUND

In this section, we provide a concise overview of how online learning, particularly regret minimization, can be employed to solve a saddle point problem. Most of the results described here can be referenced in Orabona (2019). We examine the general saddle point problem, as defined in Section 2.3.

**Definition B.1** (Saddle point problem). Let $X \subseteq \mathbb{R}^n$ and $X \subseteq \mathbb{R}^m$ and $f : X \times Y \to \mathbb{R}$. A point $(\boldsymbol{x}^\star, \boldsymbol{y}^\star) \in X \times Y$ is a saddle-point of $f$ in $X \times Y$ if

$$f(\boldsymbol{x}^\star, \boldsymbol{y}) \leq f(\boldsymbol{x}^\star, \boldsymbol{y}^\star) \leq f(\boldsymbol{x}, \boldsymbol{y}^\star), \quad \forall \boldsymbol{x} \in X, \boldsymbol{y} \in Y$$

We first describe the scenario where $f$ is a convex-concave function, meaning that $f(\cdot, \boldsymbol{y})$ is convex for any $\boldsymbol{y} \in Y$, and $f(\boldsymbol{x}, \cdot)$ is concave for any $\boldsymbol{x} \in X$. In this case, the minimax theorem v. Neumann (1928); Sion (1958) states that:

**Theorem B.2** (Minimax theorem v. Neumann (1928); Sion (1958)). *Let $X$ and $Y$ be compact, convex subsets of $\mathbb{R}^n$ and $\mathbb{R}^m$ respectively. Let $f : X \times Y \to \mathbb{R}$ a continuous, convex-concave function. Then, we have that*

$$\min_{\boldsymbol{x} \in X}\max_{\boldsymbol{y} \in Y} f(\boldsymbol{x}, \boldsymbol{y}) = \max_{\boldsymbol{y} \in Y}\min_{\boldsymbol{x} \in X} f(\boldsymbol{x}, \boldsymbol{y})$$

**Online Learning**    The saddle point problem can be reformulated as a regret minimization problem as follows. We focus on the first player, whose goal is to minimize the function $\overline{f}(\boldsymbol{x}) = \max_{\boldsymbol{y} \in \mathcal{Y}} f(\boldsymbol{x}, \boldsymbol{y})$, and Player 2, whose objective is to maximize the function $\underline{f}(\boldsymbol{y}) = \min_{\boldsymbol{x} \in \mathcal{X}} f(\boldsymbol{x}, \boldsymbol{y})$.

Then the standard suboptimality gap, that describes the improvent of Player 1 over time, is

$$\overline{f}(\boldsymbol{x}^{(t)}) - \min_{\boldsymbol{x} \in X} \overline{f}(\boldsymbol{x}) = \max_{\boldsymbol{y} \in Y} f(\boldsymbol{x}^{(t)}, \boldsymbol{y}) - \min_{\boldsymbol{x} \in X} \max_{\boldsymbol{y} \in Y} f(\boldsymbol{x}, \boldsymbol{y}) \tag{Gap$_X$}$$

Similarly, for Player 2, the corresponding measure would be:

$$\max_{\boldsymbol{y} \in X} \underline{f}(\boldsymbol{x}, \boldsymbol{y}) - \underline{f}(\boldsymbol{y}^{(t)}) = \max_{\boldsymbol{y} \in Y} \min_{\boldsymbol{x} \in X} f(\boldsymbol{x}, \boldsymbol{y}) - \min_{\boldsymbol{x} \in X} f(\boldsymbol{x}, \boldsymbol{y}^{(t)}) \tag{Gap$_Y$}$$

By adding these two measures and combining them with Sion's minimax theorem, we obtain the suboptimality (Nash) gap.

$$\max_{\boldsymbol{y} \in Y} f(\boldsymbol{x}^{(t)}, \boldsymbol{y}) - \min_{\boldsymbol{x} \in X} \max_{\boldsymbol{y} \in Y} f(\boldsymbol{x}, \boldsymbol{y}) + \max_{\boldsymbol{y} \in Y} \min_{\boldsymbol{x} \in X} f(\boldsymbol{x}, \boldsymbol{y}) - \min_{\boldsymbol{x} \in X} f(\boldsymbol{x}, \boldsymbol{y}^{(t)}) = \max_{\boldsymbol{y} \in Y} f(\boldsymbol{x}^{(t)}, \boldsymbol{y}) - \min_{\boldsymbol{x} \in X} f(\boldsymbol{x}, \boldsymbol{y}^{(t)})$$
$$\tag{Gap}$$

**Regret**    We describe the process from the perspective of Player 1. In an online learning framework, the learner chooses a strategy $\boldsymbol{x}^{(t)} \in X$ at every round $t \in [T]$. Then, the learner receives a linear loss function $\ell_X^{(t)}(\boldsymbol{x})$. Based on the feedback received thus far, they update their strategy for the next round. The standard objective is to minimize the cumulative external regret, defined as follows.

$$\mathrm{Regret}_X^{(T)}(\boldsymbol{x}) := \sum_{\tau=1}^{T} \ell_X^{(\tau)}(\boldsymbol{x}) - \sum_{\tau=1}^{T} \ell_X^{(\tau)}(\boldsymbol{x}^{(\tau)}),$$

where $t \in [T]$ is the time horizon. In other words, performance is evaluated based on the optimal fixed strategy in hindsight. To provide further precision, it's worth noting that regret is typically defined with respect to the strategy $\boldsymbol{x}$ that maximizes $\mathrm{Regret}_X^{(T)}(\boldsymbol{x})$. Nevertheless, for the sake of clarity in our explanation, we will retain the previously stated definition.

**Optimistic Regret Minimization**    Indeed, it is widely acknowledged that a wide range of online learning algorithms, including Follow-The-Regularized-Leader (FTRL) Abernethy et al. (2008); Hazan & Kale (2010) and Mirror Descent Nemirovskij & Yudin (1983), can guarantee $O(\sqrt{T})$ regret under any sequence of bounded utilities. However, when dealing with more predictable utilities, one can achieve even stronger convergence guarantees. In recent years, several studies have explored this direction Chiang et al. (2012); Rakhlin & Sridharan (2013); Syrgkanis et al. (2015). In our work, we focus on the optimistic mirror descent algorithm, as detailed in Rakhlin & Sridharan (2013).

**Optimistic Mirror Descent**    Suppose now that each player employs optimistic mirror descent. The feedback that Player 1 receives in each round is set to $\ell^{(t)}(\boldsymbol{x}) = \langle \boldsymbol{g}_X^{(t)}, \boldsymbol{x} \rangle$ where $\boldsymbol{g}_X^{(t)} = \nabla_{\boldsymbol{x}} f(\boldsymbol{x}^{(t)}, \boldsymbol{y}^{(t)})$, while Player 2 receives $\ell^{(t)}(\boldsymbol{y}) = \langle \boldsymbol{g}_Y^{(t)}, \boldsymbol{y} \rangle$ where $\boldsymbol{g}_Y^{(t)} = -\nabla_{\boldsymbol{y}} f(\boldsymbol{x}^{(t)}, \boldsymbol{y}^{(t)})$. As usual, we consider the "one-recency bias" prediction Syrgkanis et al. (2015), wherein $\tilde{\boldsymbol{g}}^{(t)} = \boldsymbol{g}^{(t-1)}$.

**Assumption B.3.** Assume that $f : X \times Y \to \mathbb{R}$ is smooth in an open interval containing its domain, in the sense that for any $\boldsymbol{x}, \boldsymbol{x}' \in X$ and any $\boldsymbol{y}, \boldsymbol{y}' \in Y$, we have the following.

$$\|\nabla_{\boldsymbol{x}} f(\boldsymbol{x}, \boldsymbol{y}) - \nabla_{\boldsymbol{x}} f(\boldsymbol{x}', \boldsymbol{y})\|_{X,*} \le L_{XX} \|\boldsymbol{x} - \boldsymbol{x}'\|_X$$
$$\|\nabla_{\boldsymbol{x}} f(\boldsymbol{x}, \boldsymbol{y}) - \nabla_{\boldsymbol{x}} f(\boldsymbol{x}, \boldsymbol{y}')\|_{X,*} \le L_{XY} \|\boldsymbol{y} - \boldsymbol{y}'\|_Y$$
$$\|\nabla_{\boldsymbol{y}} f(\boldsymbol{x}, \boldsymbol{y}) - \nabla_{\boldsymbol{y}} f(\boldsymbol{x}', \boldsymbol{y})\|_{Y,*} \le L_{XY} \|\boldsymbol{x} - \boldsymbol{x}'\|_X$$

---

**Algorithm 3:** Solving Saddle Point Problem with Optimistic OMD Orabona (2022)

---

1 **Require:** $\lambda_X > 0, \lambda_Y > 0, \boldsymbol{x}_1 \in X, \boldsymbol{y}_1 \in Y$
2 $\boldsymbol{g}_X^{(0)} = \boldsymbol{0}, \boldsymbol{g}_Y^{(0)} = \boldsymbol{0}$
3 **for** $t = 1, \cdots, T$ **do**
4 $\quad \boldsymbol{g}_X^{(t)} = \nabla_{\boldsymbol{x}} f(\boldsymbol{x}^{(t)}, \boldsymbol{y}^{(t)})$
5 $\quad \boldsymbol{g}_Y^{(t)} = -\nabla_{\boldsymbol{y}} f(\boldsymbol{x}^{(t)}, \boldsymbol{y}^{(t)})$
6 $\quad \boldsymbol{x}^{(t+1)} \in \arg\min_{\boldsymbol{x} \in X} \langle 2\boldsymbol{g}_X^{(t)} - \boldsymbol{g}_X^{(t-1)}, \boldsymbol{x} \rangle + \lambda_X B_{\psi_X}(\boldsymbol{x}; \boldsymbol{x}^{(t)})$
7 $\quad \boldsymbol{y}^{(t+1)} \in \arg\min_{\boldsymbol{y} \in Y} \langle 2\boldsymbol{g}_Y^{(t)} - \boldsymbol{g}_Y^{(t-1)}, \boldsymbol{y} \rangle + \lambda_Y B_{\psi_Y}(\boldsymbol{y}; \boldsymbol{y}^{(t)})$

8 **return** $\overline{\boldsymbol{x}}^{(T)} = \frac{1}{T} \sum_{t=1}^{T} \boldsymbol{x}^{(t)}, \overline{\boldsymbol{y}}^{(T)} = \frac{1}{T} \sum_{t=1}^{T} \boldsymbol{y}^{(t)}$

---

$$\|\nabla_{\boldsymbol{y}} f(\boldsymbol{x}, \boldsymbol{y}) - \nabla_{\boldsymbol{y}} f(\boldsymbol{x}, \boldsymbol{y}')\|_{Y,*} \leq L_{YY} \|\boldsymbol{y} - \boldsymbol{y}'\|_Y$$

where $\nabla_{\boldsymbol{x}}$ and $\nabla_{\boldsymbol{y}}$ denote the gradients with respect to $\boldsymbol{x}$ and $\boldsymbol{y}$ respectively. We denote by $(\|\cdot\|_X, \|\cdot\|)_{X,*}$ and $(\|\cdot\|_Y, \|\cdot\|)_{Y,*}$ the pair of dual norms in $X$ and $Y$.

The following theorem, originally proven in Rakhlin & Sridharan (2013), has been cited from Orabona (2022).

**Theorem B.4.** *Orabona (2022) Let $f : X \times Y \to \mathbb{R}$ a convex-concave function satisfying the Assumption B.3. For a fixed $\alpha > 0$, let $\lambda_X \geq 2\sqrt{2}(L_{XX} + L_{XY}/\alpha)$ and $\lambda_Y \geq 2\sqrt{2}(L_{YY} + L_{XY}/\alpha)$. Let $\psi_X : X \to \mathbb{R}$ be 1-strongly convex w.r.t. $\|\cdot\|_X$ and $\psi_Y : Y \to \mathbb{R}$ be 1-strongly convex w.r.t. $\|\cdot\|_Y$. Assume $\arg\max_{\boldsymbol{y} \in Y} f(\overline{\boldsymbol{x}}^{(T)}, \boldsymbol{y})$ and $\arg\min_{\boldsymbol{x} \in X} f(\boldsymbol{x}, \overline{\boldsymbol{y}}^{(T)})$ are non-empty. Then, we have*

$$\max_{\boldsymbol{y} \in Y} f(\overline{\boldsymbol{x}}^{(T)}, \boldsymbol{y}) - \min_{\boldsymbol{x} \in X} f(\boldsymbol{x}, \overline{\boldsymbol{y}}^{(T)}) \leq \frac{B_{\psi_X}(\boldsymbol{x}^{(T)}; \boldsymbol{x}^{(1)}) + B_{\psi_Y}(\boldsymbol{y}^{(T)}; \boldsymbol{y}^{(1)}) + \frac{\|\boldsymbol{g}_X^{(1)}\|_{X,*}^2}{\lambda_X} + \frac{\|\boldsymbol{g}_Y^{(1)}\|_{Y,*}^2}{\lambda_Y}}{T}$$

*where $\overline{\boldsymbol{x}}^{(T)} = \frac{1}{T} \sum_{t=1}^{T} \boldsymbol{x}^{(t)}, \overline{\boldsymbol{y}}^{(T)} = \frac{1}{T} \sum_{t=1}^{T} \boldsymbol{y}^{(t)}$.*

**Lemma B.5** (Lemma 16 Shalev-Shwartz (2007)). *$\psi(\boldsymbol{x}) = \sum_{i=1}^{n} \boldsymbol{x}_i \ln \boldsymbol{x}_i$ is 1-strongly convex with respect to the $L_1$ norm over the simplex $\Delta_n := \{\boldsymbol{x} \in \mathbb{R}^n : \boldsymbol{x}_i \geq 0, \|\boldsymbol{x}\|_1 = 1\}$.*

### B.2 PROOF OF LEMMA 3.6

In the proof of the lemma, we consider the function $f_\lambda(\boldsymbol{x}, \boldsymbol{y}) = \boldsymbol{x}^\top (\mathbf{A} - \mathbb{1}\lambda\boldsymbol{b}^\top)\boldsymbol{y} - \frac{1}{2}(\boldsymbol{x}^\top \boldsymbol{a} - \lambda)^2$, where $\boldsymbol{x}$ is the maximizer and $\boldsymbol{y}$ is the minimizer in our context, and we define $X := \Delta_n$ and $Y := \Delta_m$. Due to this, we select the negative entropy as the regularizer for both $X$ and $Y$, that is, $\psi_X(\boldsymbol{x}) = \sum_{i=1}^{n} \boldsymbol{x}_i \ln \boldsymbol{x}_i$. According to Lemma B.5, this choice ensures that the regularizer is 1-strongly convex concerning the $L_1$ norm, satisfying some of the conditions outlined in Theorem B.4.

Next, we examine whether Assumption B.3 are satisfied. Instead of deriving each constant individually, we utilize Proposition B.6. The Hessian matrix of $f_\lambda$ is a block matrix composed of four blocks, $\boldsymbol{H}_{f_\lambda} = \begin{pmatrix} \boldsymbol{a}\boldsymbol{a}^\top & \mathbf{A} \\ \mathbf{A}^\top & \mathbf{0} \end{pmatrix}$. Therefore, we can readily bound each constant $L_{XX}, L_{XY}, L_{YY}$ by $4 \max\{\|\mathbf{A}\|_2, \|\boldsymbol{a}\|_2\}$.

**Proposition B.6** (Claim C.2 Leonardos et al. (2021)). *Consider a symmetric block matrix $C$ with $n \times n$ matrices so that $\|C_{ij}\|_2 \leq L$. Then, it holds that $\|C\|_2 \leq nL$. In other words, if all block matrices have spectral norm at most $L$, then $C$ has spectral norm at most $nL$.*

Regarding the initial Bregman divergences $B_{\psi_X}(\boldsymbol{x}^{(T)}; \boldsymbol{x}^{(1)})$ and $B_{\psi_Y}(\boldsymbol{y}^{(T)}; \boldsymbol{y}^{(1)})$, it's worth noting that when $\boldsymbol{x}^{(1)}$ and $\boldsymbol{y}^{(1)}$ are initialized to the uniform distribution, they are both upper-bounded by $\ln n$ and $\ln m$, respectively:

$$B_{\psi_X}(\boldsymbol{u}; \boldsymbol{x}) = \sum_{i=1}^{n} \boldsymbol{u}_i \ln \boldsymbol{u}_i + \ln n \leq \ln n \quad \text{and} \quad B_{\psi_Y}(\boldsymbol{u}; \boldsymbol{y}) = \sum_{j=1}^{m} \boldsymbol{u}_j \ln \boldsymbol{u}_j + \ln m \leq \ln m$$

Lastly, we will bound the dual norm of the gradients vector. For the sake of completeness, we will present the upper bound for $\|\boldsymbol{g}_X^{(1)}\|_{X,*}^2$.

$$\|\boldsymbol{g}_X^{(1)}\|_{X,*} = \|\boldsymbol{g}_X^{(1)}\|_\infty \leq \|(\mathbf{A} - \mathbb{1}\lambda\boldsymbol{b}^\top)\boldsymbol{y}\|_\infty + |\boldsymbol{x}^\top\boldsymbol{a} - \lambda|\|\boldsymbol{a}\|_\infty$$
$$\leq \max_{i \in [n]}\{(\mathbf{A} - \mathbb{1}\lambda\boldsymbol{b}^\top)_i\boldsymbol{y}\} + 3\|\boldsymbol{a}\|_\infty \leq (\mathbf{A}_{\max} + 1) + 3$$
$$\|\boldsymbol{g}_Y^{(1)}\|_{Y,*} = \|\boldsymbol{g}_Y^{(1)}\|_\infty \leq \|(\mathbf{A} - \mathbb{1}\lambda\boldsymbol{b}^\top)^\top\boldsymbol{x}\|_\infty \leq (\mathbf{A}_{\max} + 1)$$

**Lemma B.7** (Proof of Lemma 3.6). *Let $(\mathbf{x}^{(0)}, \mathbf{y}^{(0)})$ be initialized as uniform vectors $(\mathbb{1}/n, \mathbb{1}/m)$. Suppose that both players employ OMWU for $T = \Omega\left(\frac{(\ln n + \ln m)}{\epsilon}\right)$ with learning rates $\eta_X = \eta_Y = \frac{1}{16\sqrt{2}\max\{\|\mathbf{A}\|_2, \|\boldsymbol{a}\|_2\}}$ and fixed $\lambda$. Then, Algorithm 2 returns a pair of strategies $\left(\overline{\boldsymbol{x}}^{(T)}, \overline{\boldsymbol{y}}^{(T)}\right)$ that guarantees the following.*

$$Gap\left(\overline{\boldsymbol{x}}^{(T)}, \overline{\boldsymbol{y}}^{(T)}\right) = \max_{\boldsymbol{x} \in \Delta_n} f_\lambda\left(\boldsymbol{x}, \overline{\boldsymbol{y}}^{(T)}\right) - \min_{\boldsymbol{y} \in \Delta_m} f_\lambda\left(\overline{\boldsymbol{x}}^{(T)}, \boldsymbol{y}\right) \leq \epsilon.$$

*Also, it holds that $\left(\overline{\boldsymbol{x}}^{(T)}, \overline{\boldsymbol{y}}^{(T)}\right)$ is an $\epsilon$-approximate stationary point of $f_\lambda$.*

*Proof.* As mentioned previously, all the assumptions and required conditions of Theorem B.4 have been satisfied. By setting $\alpha = 1$, $\lambda_X = \lambda_Y = 16\sqrt{2}\max\{\|\mathbf{A}\|_2, \|\boldsymbol{a}\|_2\}$, bounding $\|\boldsymbol{g}_X^{(1)}\|_{X,\infty}^2$, $\|\boldsymbol{g}_Y^{(1)}\|_{Y,\infty}^2$ by $(\mathbf{A}_{\max} + 4)$ and $T \geq O(\frac{\ln n + \ln m}{\epsilon})$, we obtain the following results from Theorem B.4.

$$\max_{\boldsymbol{x} \in \Delta_n} f_\lambda(\boldsymbol{x}, \overline{\boldsymbol{y}}^{(T)}) - \min_{\boldsymbol{y} \in \Delta_m} f_\lambda(\boldsymbol{x}, \overline{\boldsymbol{y}}^{(T)}) \leq \epsilon$$

The fact that $(\overline{\boldsymbol{x}}^{(T)}, \overline{\boldsymbol{y}}^{(T)})$ is an $\epsilon$-stationary point can be deduced from the definition of the Nash Gap, the non-negativity of both terms, and the definition of a convex-concave function for $f_\lambda$. □

## C EXPERIMENTS

In this section, we provide experimental evaluation supporting our theoretical finding. Here we restrict to a single example, while we also provide an anonymous repository containing multiple rank-1 games of multiple sizes

$$\mathbf{A} = \begin{bmatrix} -0.10 & 0.50 & 0.50 & -0.00 & -0.30 \\ 0.00 & -0.10 & -0.30 & 0.10 & -0.20 \\ -0.10 & -0.20 & -0.30 & 0.00 & -0.30 \\ -0.30 & 0.20 & -0.10 & -0.20 & 0.00 \\ -0.40 & -0.20 & -0.30 & 0.40 & -0.30 \end{bmatrix} \quad \boldsymbol{a} = \begin{bmatrix} 0.60 \\ 0.80 \\ 0.80 \\ 0.00 \\ 0.10 \end{bmatrix} \quad \boldsymbol{b} = \begin{bmatrix} 0.30 \\ 1.00 \\ 0.50 \\ 0.60 \\ 0.10 \end{bmatrix}$$

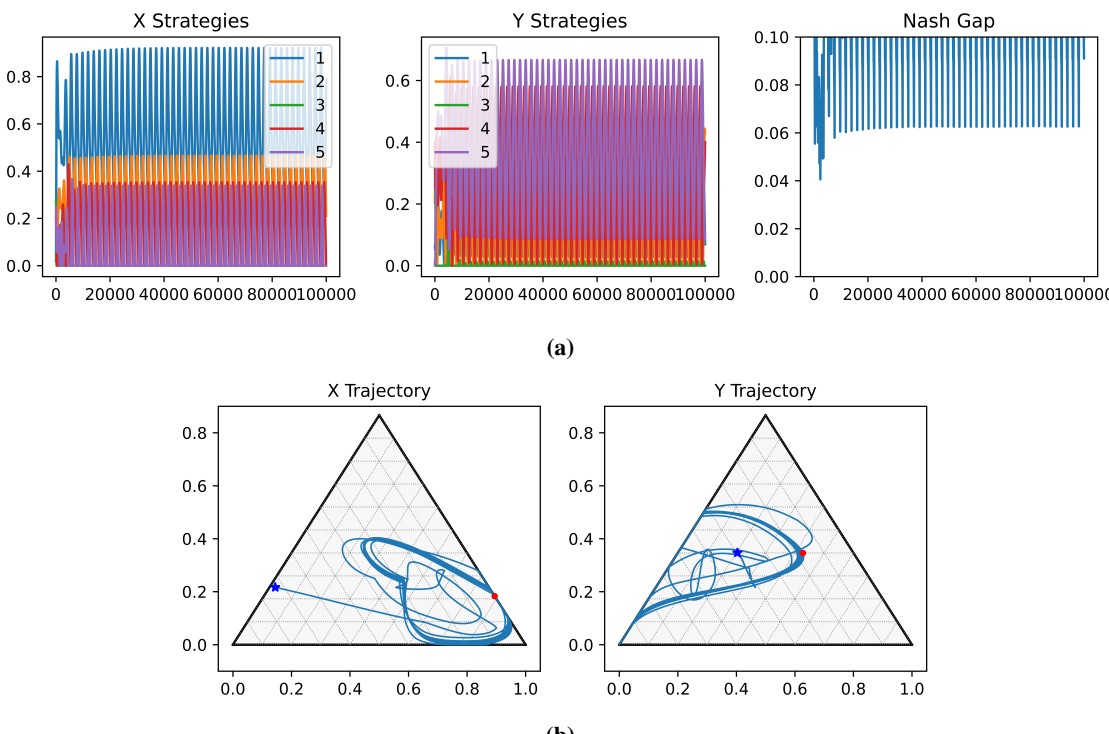

**(a)**

**(b)**

**Figure 1:** In Figure 1a, we observe how the strategies of the players evolve over time when they employ vanilla OGA, alongside with the Nash gap–as defined in (Gap). In Figure 1b, we plot the respective trajectories on the simplex signifying that OGA enter a limit cycle.

We constructed a *random* $6 \times 6$ game defined by the matrix $\mathbf{A}$, and the vectors $\boldsymbol{a}, \boldsymbol{b}$. Then, the payoff matrix of the second player is derived as usual: $\mathbf{B} = -\mathbf{A} + \boldsymbol{a}\boldsymbol{b}^\top$. Our solution delineated in Section 3.2 is compared against a (vanilla) optimistic variant of mirror descent. More specifically, our tests were performed with (optimistc) gradient ascent (OGA) instead of multiplicative weights to avoid any numerical imprecision that might arise due to exponentiation.

In Figure 1, we (experimentally) verify that OGA enters a limit cycle, whereas our solution Figure 2 reaches a Nash equilibrium. While this game may not have a natural interpretation initially, given their random construction, it is worth noting that constructing random games where OGA provably enters a limit cycle is rather challenging. This difficulty is further explained by the recent work Anagnostides et al. (2022a).

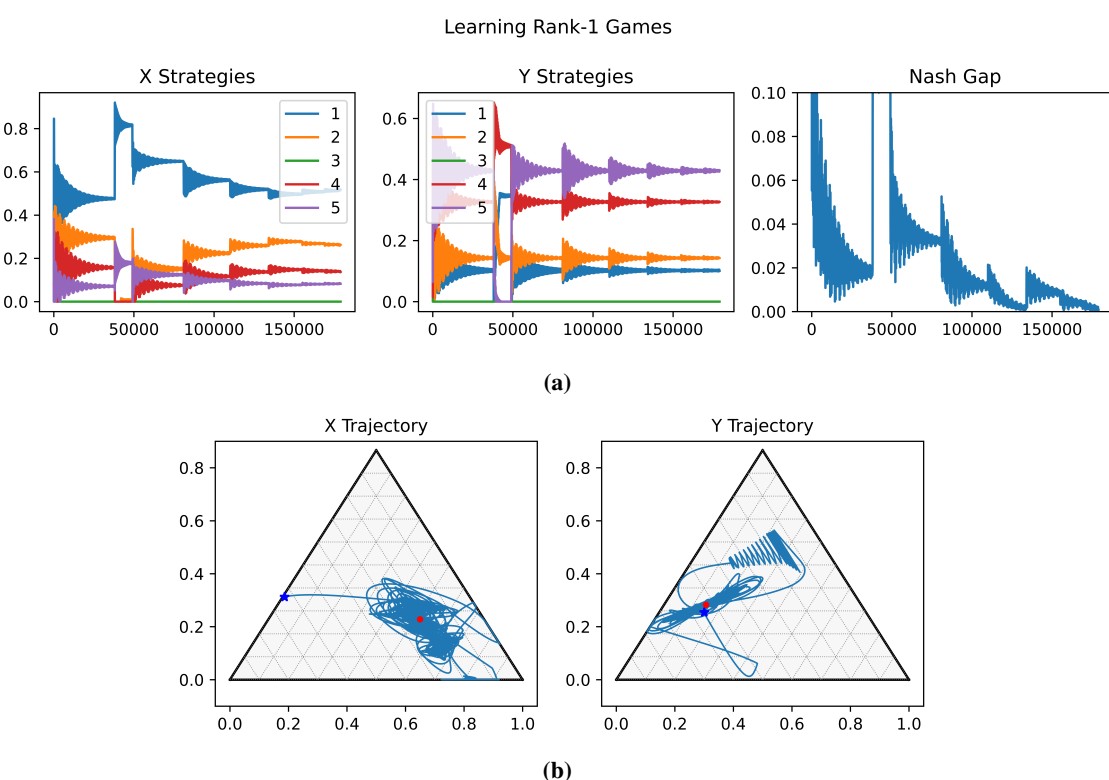

**Figure 2:** In Figure 2a, we observe how the strategies of the players evolve over time when they employ Algorithm 1, alongside with the Nash gap–as defined in (Gap). In Figure 2b, we plot the respective trajectories on the simplex as well.

