# OpenReview forum: "Learning Nash Equilibria in Rank-1 Games"
_ICLR.cc/2024/Conference — ICLR 2024 poster_

### Official Review · Reviewer_LH17 · 2023-10-30

**Soundness:** 3 good
**Presentation:** 2 fair
**Contribution:** 3 good
**Rating:** 5
**Confidence:** 2

**Summary:**

The authors consider the problem of approximating the Nash equilibrium (NE) of a rank-one two-player game where the sum of the payoff matrices is of rank one. They prove that a modification of the optimistic mirror descent converges, used to learn a NE in zeros-sum (rank-0) games, to an \epsilon-approximate NE after O(1/\epsilon^2\log(1/\epsilon)) iterations. They achieves this by leveraging a reduction of rank-1 games to rank-0 games building on the results by Adsul et al. (2021).

**Strengths:**

I think the proposed algorithm and associated analysis is a natural and valuable contribution. The proofs seem correct and are easy to read as far as I checked.

**Weaknesses:**

The general presentation could be improved, see specific and general comments. The literature review could also be improved in particular it is not clear what is the exact contribution of this work   in comparison to the one by Adsul et al. (2021).

**Questions:**

#General comments:

- Do you have a concrete/practical example of a rank-one game?

- It would be interesting to provide some preliminary experiments on some toy games to compare the proposed algorithm to vanilla OMWU.

- The intuition at the end of Section 1 is not very clear in particular about the fundamental connections between rank-one and rank-zero games.

- It would be clearer to introduce  definitions on the equivalence of games since many technical points rely on jumping from on formulation to another.

- How do your algorithm compare with the one of Adsul et al. (2021) in terms of rates, complexity; since both algorithms seem to solve the same problems with similar techniques.

- What can we say about rank-two games?

#Specific comments:

- P1, introduction: it is not clear what is the MVI property. And "games, We".

- P2, top: It is still not clear what is the Minty property. Maybe you can compare the rate you obtain with the rate one can obtain in a rank-0 game. Is \lambda a real parameter or also a vector? Do we really have that extra gradient and optimistic mirror descent fails to converge if the Minty property is not verified or we can just say we cannot provide any guaranties in this case?

- P3, Lemma 2.6:  what do you mean exactly by a game "can be written" in a certain form ?

- P3, end: "In the what follows"

- P5, end of Section 2.3.1: It is still not clear what is the Minty criterion.  The link between (MVI) and (VI) and when it is possible to solve (VI) with extra-gradient of optimistic mirror descent is also not clear, e.g. what do you mean by "the Minty variational inequality is satisfied'. You also talk at some point about saddle point in the context of VI. Maybe you could specialized the results for saddle point without introducing the setting of variationnal inequalities.

- P7, Algorithm 1: How do you initialize \lambda?

- P9, before Lemma 3.6: How do you initialize  \lambda, and maybe you should precise in
which interval \lambda lies.

- P13, before (8): Can you detail the second inequality. I think  you should work with the difference \lambda-xa and upper bound this difference after applying Cauchy-Schwartz inequality.

---

> ### Author Response · Authors · 2023-11-19
> **Response to Reviewer LH17**
>
> We appreciate the reviewer's helpful comments. In response to the length restriction, we have divided our reply into two parts.
>
> > Do you have a concrete/practical example of a rank-one game?
>
> As stated in the paper by Kannan and Theobald [2010], where the rank hierarchy of bimatrix games was originally proposed, rank-1 games resembles multiplication games introduced in \Bulow and Levin [2006]. In these games, $n$ firms and $n$ workers are matched together, and the value of the match is determined by a multiplication. However, it is a game involving $2n$ players rather than two. For this reason, Adsul et al. [2018] proposed a simple "trading game" as a starting point for further investigation into an economic interpretation of this class of games. Nevertheless, we believe that the main merit of our work stems from a theoretical standpoint. Learning Nash equilibria in bimatrix games beyond zero-sum is a crucial stepping stone for either investigating efficient methods for games of higher rank or introducing similar hierarchies over games that may characterize more precisely the difficulty of each class in this hierarchy. Additionally, as shown in Section 3.1, existing methods for monotone settings (convex-concave optimization) and beyond (including those satisfying (weak) Minty conditions) prove ineffective in rank-1 games. Hence, we believe that the optimization problem presented in Lemma 3.4 holds promise for applications in addressing higher-rank games.
>
> > It would be interesting to provide some preliminary experiments on some toy games [...]
>
> We appreciate the reviewer for suggesting the idea of experimental evaluation; in response, we posted a general comment. Notably, our tests were performed with gradient ascent instead of multiplicative weights to avoid any numerical imprecision that might arise due to exponentiation.
>
> Both algorithms, OGA and OMWU, share similar guarantees in terms of the approximation error, differing only in their dependency on the number of strategies. OGA has a $\sqrt{n}$ dependency, while OMWU has a $\log(n)$ dependency. It's important to note that both algorithms are quite similar, being byproducts of the (meta) algorithmic paradigm: online mirror descent.
>
> We generated four random rank-1 games of various sizes where OGA enters a limit cycle. In contrast, our solution reaches a Nash equilibrium. While these games may not have a natural interpretation initially, given their random construction, it's worth noting that constructing random games where OGA provably enters a limit cycle is rather challenging. This difficulty is further explained by the recent work in
>
> > The intuition at the end of Section 1 [...] the fundamental connections between rank-one and rank-zero games.
>
> In their work, Kannan and Theobald [2010] introduced a hierarchy of two-player bimatrix games $(A,B)$ based on the rank of the matrix $A+B$. This hierarchy essentially separates the general class of bimatrix games into groups with the same rank. To the best of our knowledge, there is no apparent connection between games with different ranks. We presume that the original idea in Kannan and Theobald [2010] also inspired by the work in Lipton et al. [2003], was to interpret rank as an indicator of the difficulty level of a game; the higher the rank, the more challenging it is to solve. For example, in rank 0 classes, the payoff matrices necessarily satisfy $A+B=0 \Rightarrow B=-A$; these are the zero-sum games for which was already well-known that are efficiently solved.
>
> >  It would be clearer to introduce definitions on the equivalence of games [...]
>
> We will ensure to refine the introductory part, providing a more comprehensive explanation of the motivation behind the rank games hierarchy, as described above.
>
> > "How do your algorithm compare with the one of Adsul et al. (2021) in terms of rates, complexity [...]
>
> While the core concept of reducing a rank-1 game to a parameterized zero-sum is rooted in Adsul et al. [2018], our approach diverges fundamentally in some aspects. Our primary focus is on learning Nash equilibria. In Lemma 3.3, we establish a novel connection between the approximate Nash equilibrium of the rank-1 game and those in a parameterized zero-sum, a previously unknown relationship. Building on the insights from Lemma 3.3, we introduce the function $f_{\lambda}$, and, as proven in Lemma 3.4, the stationary points of this function can reliably be extended to Nash equilibrium points of the parameterized zero-sum game and, consequently, to those of rank-1 games.
> In contrast, the approach in Adsul et al. [2018] is solely based on characterizing the optimal faces of the best response polytopes, as proven in Adler and Monteiro [1992]. This approach necessitates solving up to four linear programs for each value of $\lambda$, and despite its polynomial running time in the description of the program, it takes considerably more time.

---

> ### Author Response · Authors · 2023-11-19
> **Response to Reviewer LH17 (Part 2)**
>
> > What can we say about rank-two games?
>
> Notwithstanding the above, the question you have raised is highly significant and remains an open problem, as emphasized in Daskalakis [2018]. While the computational complexity of finding the \textit{exact} Nash equilibrium in rank-3 games has been settled to be PPAD-hard, the status of rank-2 games is still undetermined. In Adsul et al. [2018], the authors assert that rank-2 games are also PPAD-hard, citing "personal communication," which, as of now, remains unpublished.
>
> **Specific Comments**
>
> > P1, introduction: it is not clear what is the MVI property. And "games, We".
>
> We will further improve Section 2.3 by providing a more complete description of variational inequalities (VI) and addressing specifically the case of Minty VI. In essence, many optimization problems can be reformulated into an equivalent VI form. However, in a nonconvex-nonconcave setting (where the operator in the variational inequality is not monotone), several lines of work exist. This includes extensions of the min-max theorem beyond convex-concave objectives, with a notable example being settings that satisfy the Minty VI Mertikopoulos et al. [2018]. There are also investigations into weak versions of it, as seen in Diakonikolas et al. [2021]. In the revised version, we will ensure that these points are more clearly stated. Thank you for catching the typo as well.
>
> > P2, top: It is still not clear what is the Minty property. Maybe you can compare the rate you obtain with the rate one can obtain in a rank-0 game. Is $\lambda$ a real parameter or also a vector? Do we really have that extra gradient and optimistic mirror descent fails to converge if the Minty property is not verified or we can just say we cannot provide any guaranties in this case?
>
> We have addressed the concern on Minty VI in the question above. For the case of rank-1 games, also known as zero-sum, and when working in the context of learning, a list of best-known rates can be found in Daskalakis et al. [2021]. However, if our interest is in computing a Nash equilibrium, we can achieve that using linear programming.
>
> > P7, Algorithm 1: How do you initialize \lambda?
>
> > P9, before Lemma 3.6: How do you initialize \lambda, and maybe you should precise in which interval \lambda lies.
>
> Regarding $\lambda$, it is a scalar, and we note that it is a convex combination of the elements of the vector $\alpha$. Now, the elements of $\alpha$ are confined within the interval $[-1,1]$ without loss of generality$^{1}$, and so it follows that $\lambda$ is in $[-1,1]$. As discussed in a previous question, satisfying the (weak) Minty VI is a sufficient condition for optimistic mirror descent (a variant of extra gradient, respectively) to converge in settings beyond convex-concave. Thus, as you pointed out in cases where this does not hold, we cannot provide any guarantees.
>
> > P3, Lemma 2.6: what do you mean exactly by a game "can be written" in a certain form ?
>
> As this lemma is not part of our work, we presented it as originally stated in Adsul et al. [2018]. It states that a game $(A,B)$, where $\texttt{rank}(A+B)=r$, can also be expressed as $(A,C+ ab^{\top})$, given that $\texttt{rank}(A+C)=r-1$ and $\texttt{rank}(a b^{\top})=1$.
>
> > P3, end: "In the what follows"
>
> That is a typo, thanks for spotting that. We will correct it in the revised version.
>
> > P13, before (8): Can you detail the second inequality.
>
> If we are mistaken, you are referring to the inequality after that of an approximate Nash equilibrium. This follows from the fact that $x$ (and $y$) is a probability distribution and so $\mathbf{1}^{\top} x = 1$. If we have not addressed your comment correctly, please write us below and will response afterwards.
>
> $^1$ That is why Nash equilibria remain invariant under the transformations of shifting and scaling of the payoffs.
>
> 1. R. Kannan and T. Theobald. Games of fixed rank: A hierarchy of bimatrix games.
>
> 2. J. Bulow and J. Levin. Matching and price competition.
>
> 3. B. Adsul, J. Garg, R. Mehta, M. Sohoni, and B. V. Stengel. Fast algorithms for rank-1 bimatrix games.
>
> 4. I. Anagnostides, G. Farina, I. Panageas, and T. Sandholm. Optimistic mirror descent either converges to nash or to strong coarse correlated equilibria in bimatrix games.
>
> 5. R. J. Lipton, E. Markakis, and A. Mehta. Playing large games using simple strategies.
>
> 6. I. Adler and R. D. Monteiro. A geometric view of parametric linear programming.
>
> 7. C. Daskalakis. Equilibria, fixed points, and computational complexity-nevanlinna prize lecture.
>
> 8. P. Mertikopoulos, B. Lecouat, H. Zenati, C.-S. Foo, V. Chandrasekhar, and G. Piliouras. Optimistic mirror descent in saddle-point problems.
>
> 9. J. Diakonikolas, C. Daskalakis, and M. I. Jordan. Efficient methods for structured nonconvex-nonconcave min-max optimization.
>
> 10. C. Daskalakis, M. Fishelson, and N. Golowich. Near-optimal no-regret learning in general games.

---

> > ### Author Response · Authors · 2023-11-21
> > **Have the concerns raised by the reviewer been sufficiently addressed?**
> >
> > We thank again the reviewer for the helpful feedback. Given that the discussion period is soon coming at an end, please let us know if we have adequately addressed the concerns raised, and if the reviewer has any further questions.

---

### Official Review · Reviewer_r4eu · 2023-10-31

**Soundness:** 3 good
**Presentation:** 4 excellent
**Contribution:** 2 fair
**Rating:** 6
**Confidence:** 4

**Summary:**

This paper studies the problem of learning Nash equilibrium in two-player rank-1 games, which does not satisfies the Minty condition. The main contribution of the paper is an efficient algorithm for learning Nash equilibrium in two-player zero-sum games. The main technical insights that enables the result are structural results of rank-1 games. Specifically, it is shown that the set of Nash equilibria of a rank-1 game is equivalent to the set of Nash equilibria of a two-player zero-sum game (with additional constraints, so can be seen as a saddle-point problem). The proposed algorithm is double-loop in the sense that the inner loop runs OWMU on the parameterized zero-sum, finds an approximate Nash equilibrium and the outer loop updates the parameter.

**Strengths:**

This paper is fairly well-written and easy-to-follow. I really appreciate the authors for providing very detailed high-level iades and proofs for every technical results, which makes the paper very clear.

**Weaknesses:**

Several concerns about the technical contributions.

1. There is no lower bound results and it is not know if the provided convergence rate is optimal.
2. The authors show that there exists a rank-1 game that does not satisfies the Minty condition, so previous results on efficient convergence does not apply directly to rank-1 games. However, recent results have shown efficient convergence under a weaker condition than the Minty condition, called *weak Minty* [1, 2, 3].  [1] proposed the notion of weak MVI, while [2,3] generalized the notion to the constrained setting.
3. I am not sure whether Algorithm 1 is a decentralized algorithm since it is a double-loop algorithm and requires some coordination between the two players. Thus Algorithm 1 is more like an algorithm that computes the Nash equilibrium, not a decentralized learning dynmics that converges to Nash equilibrium

[1] Efficient Methods for Structured Nonconvex-Nonconcave Min-Max Optimization, Diakonikolas et al., AISTATS 2021

[2] Escaping limit cycles: Global convergence for constrained nonconvex-nonconcave minimax problems, Pethick et al., ICLR 2022

[3] Accelerated Single-Call Methods for Constrained Min-Max Optimization, Cai and Zheng, ICLR 2023

**Questions:**

1. Does rank-1 games satisfy the weak Minty condition?
2. Can the authors comment more on Algorithm 1? Is it possible to get a single-loop and decentralized algorithm?

---

> ### Author Response · Authors · 2023-11-19
> **Response to Reviewer r4eu**
>
> We thank the reviewer for the helpful feedback. We address their comments in the order they were stated.
>
> 1. The reviewer highlights an important consideration. Given that our approach hinges on the idea of reducing a rank-1 game into a zero-sum game and then into a carefully crafted minmax optimization problem in Section 3.2, we consider algorithms revolving around those settings. Regarding first-order methods, there is a lower bound of $\Omega(\frac{1}{T})$, see Daskalakis et al. [2011], making it impossible to achieve faster convergence in zero-sum games without using some sort of coordination or higher-order methods. However, the apparent gap in our solution, as the running time depends on $O(\frac{1}{\epsilon^2})$, comes from the regularization term. Thus, unless an entirely different methodology or accelerated convergence methods, which may not align well with the learning in games setting, are explored, we assert that our proposed solution remains optimal.
>
> 2. We thank reviewer for highlighting the case of weak MVI, as we had not originally considered it in our work. Unfortunately, even weak Minty fails to hold in our case. The simplest way to verify this is through the same example we presented in Section 3.1 for the case of MVI. Following the definition in Diakonikolas et al. [2021], weak Minty states the following:
>
>
> $ \langle F(z), z - z^{\star} \rangle \geq - \frac{\rho}{2}  \lVert F(z) \rVert_2^2  \quad  \text{for } \rho \in \left[0, \frac{1}{4L} \right) $
> where $L$ is the Lipschitz constant of the operator $F(z) = \begin{bmatrix} -Ay & -B^{\top} x   \end{bmatrix}^{\top}$.
>
> Although in Pethick et al. [2022] the weak minty has a different dependency in $\rho$; specifically, weak MVI states the following.
>
> $$
> \begin{align}
> \langle F(z), z - z^{\star} \rangle
> \geq
> \rho
> \lVert F(z) \rVert_2^2
> \quad
> \text{for } \rho \in \left( - \frac{1}{2L}, \infty \right)
> \end{align}
> $$
>
> To calculate the Lipschitz constant, we have to calculate the spectral norm of the block matrix $
> C =
> \left[
> \begin{smallmatrix}
> \mathbf{0} & -A \\\\
> -B^{\top} & \mathbf{0}
> \end{smallmatrix}
> \right]
> $; that is, $L = \lVert C \rVert_2 = 2.288$.
>
> Let $z_1 = (x_1,y_1), z_2 = (x_2,y_2), z_3 = (x_3,y_3)$ be the Nash equilibria of Example 3.2. Then, its relative straightforward to show the following.
>
> $$
> \begin{align}
> &&
> \lVert F(z_1) \rVert_2^2 = 6, \
> \lVert F(z_2) \rVert_2^2 = 2, \
> \lVert F(z_3) \rVert_2^2 = 1
> &&
> \\\\
> &&
> \langle F(z_2), z_2 - z_1  \rangle = -2, \
> \langle F(z_1), z_1 - z_2 \rangle = -4, \
> \langle F(z_1), z_1 - z_3 \rangle= -9/4
> &&
> \end{align}
> $$
>
> We can easily verify that the weak MVI is not satisfied for neither of those definition. We will expand our example in the Appendix including the case of weak MVI.
>
> 3. We mentioned here parts of our response to Reviewer TKWX review as well. First, we would like to emphasize that our seemingly double-loop algorithm can indeed be expressed as a single loop as follows: there is a single loop of K steps $K=  \frac{\log n + \log m}{\epsilon^2} \log \frac{1}{\epsilon} $ and $\lambda$ is updated after every $\frac{\log n + \log m}{\epsilon^2}$ steps using the same rule. Concerning the decentralized nature of our approach, the agents are not permitted to coordinate-implying a lack of communication between them—and and their strategy updates are based solely on first-order queries; for these reasons, we characterize our approach as having uncoupled/decentralized dynamics.
>
> **Question 1**: Does rank-1 games satisfy the weak Minty condition?
>
>  As we show in Item 2 above, weak MVI is not satisfied in the setting we consider.
>
> **Question 2** : Can the authors comment more on Algorithm 1? Is it possible to get a single-loop and decentralized algorithm?
>
> Yes, we can in fact achieve that pretty straightforwardly. We specifically explained the process in Item 3 above.
>
> 1. C. Daskalakis, A. Deckelbaum, and A. Kim. Near-optimal no-regret algorithms for zero-sum games.
>
> 2. J. Diakonikolas, C. Daskalakis, and M. I. Jordan. Efficient methods for structured nonconvex-nonconcave min-max optimization.
>
> 3. T. Pethick, P. Latafat, P. Patrinos, O. Fercoq, and V. Cevher. Escaping limit cycles: Global convergence for constrained nonconvex-nonconcave minimax problems.

---

> > ### Comment · Reviewer_r4eu · 2023-11-21
> > **Example on Weak MVI**
> >
> > I thank the authors for their response.
> >
> > I have one question on the provided example. In the example, all the quantities $\langle F(z), z - z^*\rangle$ calculated are positive, thus clearly satisfy the (weak) MVI condition. Am I mising something?

---

> ### Author Response · Authors · 2023-11-21
> **Reply on reviewer's comment**
>
> We appreciate the reviewer's response.
>
> Apologies for any confusion – we accidentally omitted the minus sign; we have updated our original comment as well. Additionally, we want to point out that our choice of $z$ and $z^{\star}$ as Nash Equilibria implies that the inner products are inherently non-positive, and as such in Example 3.2, they are bounded away from $0$, not just by a small margin.
>
> Please let us know if we have adequately addressed the concerns raised, and if the reviewer has any further questions.

---

> > ### Comment · Reviewer_r4eu · 2023-11-21
> >
> > Thanks for your clarification.
> >
> > I think this paper provides interesting results on learning in rank-1 games,  which is a class of non-monotone games. I have increased my score but I still think it is a weakness that there is no matching lower bound results.

---

> > > ### Author Response · Authors · 2023-11-22
> > >
> > > We thank the reviewer for increasing their score after our provided feedback. We agree that is an interesting open question whether the convergence rate of our learning algorithm is optimal or can be improved, though the scope of our paper is to provide convergence results for learning Nash equilibria for games that do not satisfy the (weak) minty property (which was the state-of-the-art for games). We do hope and believe that our positive results will motivate further the investigation of other game settings in which learning Nash equilibrium is possible.

---

### Official Review · Reviewer_TKWX · 2023-11-01

**Soundness:** 3 good
**Presentation:** 3 good
**Contribution:** 3 good
**Rating:** 8
**Confidence:** 3

**Summary:**

The authors study the problem of finding Nash equilibria for bimatrix rank-1 games. They propose a novel decentralized algorithm for finding approximate Nash equilibria of rank-1 games. Their approach works by reducing the problem to a sequence of parametric zero sum games where the parameter changes for each step of the sequence. Approximately solving the zero sum game and then adapting the game parameter repeatedly leads to an approximate Nash equilibrium for the original game. This decomposition allows the authors to bypass the fact that the original game does not satisfy the Minty variational inequality.

**Strengths:**

To the best of my knowledge, this work is the first to propose gradient based decentralized algorithms for approximate Nash equilibria of rank one games. Even though the general solution framework is the same as in Adsul et al. 2021, as they both reduce the problem to a sequence of parametric zero sum games, the way each zero sum game is tackled in this work is different and novel. Handling the additional constraint to the zero sum game was non-trivial. In addition, showcasing new  decentralized learning algorithms for games that go beyond the MVI property may be of independent interest.

**Weaknesses:**

The algorithm proposed is decentralized in a somewhat limited sense. For example, the two players need to coordinate to solve the same parametric game zero sum in each iteration.

**Questions:**

I think some more explanation about how we computed the bound on K would help. I guess I am missing the argument about why we only need to know $\lambda$ up to $O(\epsilon)$ accuracy or at least some similar argument for the binary search termination.

---

> ### Author Response · Authors · 2023-11-19
> **Response to Reviewer TKWX**
>
> We sincerely appreciate Reviewer TKWX's positive feedback on our proposed solution. We acknowledge their concern regarding the decentralized nature of our approach and would like to provide further clarification.
>
> Given that agents are not permitted to coordinate-implying a lack of communication between them—and considering that their strategy updates are based solely on first-order queries, we characterize this as an uncoupled/decentralized dynamics.
> If the reviewer's point of confusion stems from the apparent dissimilarity in the update mechanism between the strategies and the parameter $\lambda$ we would like to provide additional clarity on this aspect. In contrast to the strategies, which are updated at every timestep, the parameter $\lambda$ follows a periodic update pattern that serves a specific purpose: to accurately infer whether the correct stationary point of $f_{\lambda}$ has been reached, or if $\lambda$ should be adjusted to minimize the difference $|x^{\top} \alpha - \lambda|$.
>
> Our proposed setting can also be put in the recent context of time-varying games, where we allow the game to change over time, even adversially as proven in Zhang et al. [2022], Anagnostides et al. [2023]. Hence, a compelling future question, arising from both your review and the response from Reviewer r4eu, is whether it is feasible to reinterpret the parameterized zero-sum as a time-varying zero-sum and employ those techniques in the analysis.
>
> **Question** "I think some more explanation about how we computed the bound on K would help. ..."
>
> We acknowledge the need for additional clarification for $K$. To address this, we begin by noting that $\lambda$ is a convex combination of the elements of vector $\alpha$. Now, the elements of $\alpha$ are confined within the interval $[-1,1]$ without loss of generality$^1$, and so it follows that $\lambda$ is in $[-1,1]$. Thus, considering a correct value $\lambda^{\star}$-i.e., corresponding to a Nash equilibrium $(x^{\star}, y^{\star})$ of the rank-1 game where $(x^{\star})^{\top} \alpha = \lambda^{\star}$-we notice that an interval of length $O(\epsilon)$ around $\lambda^{\star}$, contains acceptable values of $\lambda$ as it can shown by Lemma 3.3. Consequently, by partitioning $[-1,1]$ into intervals of length $\epsilon$, a binary search will require at most $2 \log(1/\epsilon)$ steps. We will definitely revise our manuscript to clarify the point Reviewer TKWX raised.
>
> $^1$ That is why Nash equilibria remain invariant under the transformations of shifting and scaling of the payoffs.
>
> 1. M. Zhang, P. Zhao, H. Luo, and Z. Zhou. No-regret learning in time-varying zero-sum games.
>
> 2. I. Anagnostides, I. Panageas, G. Farina, and T. Sandholm. On the convergence of noregret learning dynamics in time-varying games.

---

> > ### Author Response · Authors · 2023-11-21
> > **Have the concerns raised by the reviewer been sufficiently addressed?**
> >
> > We thank again the reviewer for the helpful feedback. Given that the discussion period is soon coming at an end, please let us know if we have adequately addressed the concerns raised, and if the reviewer has any further questions.

---

> > > ### Comment · Reviewer_TKWX · 2023-11-22
> > > **Concerns addressed**
> > >
> > > I would like to thank the authors for their response. All of my questions/concerns have been addressed!

---

### Official Review · Reviewer_cjaZ · 2023-11-10

**Soundness:** 3 good
**Presentation:** 2 fair
**Contribution:** 1 poor
**Rating:** 5
**Confidence:** 4

**Summary:**

The paper focuses on learning (approximate) Nash Equilibria of rank-1 bimatrix game, which can comprise a potentially large number of disconnected components, and may lack convexity in general.
The rank-1 bimatrix games, which have been shown by Adsul et al., 2021  to have polynomial-time algorithms, are especially interesting, as k>=3 are PPAD-hard, and the complexity of k=2 games are left as an open question.

The authors build upon the reparameterization approach of Adsul et al., 2021, which establishes a link between rank-1 games and parameterized zero-sum games.

- In Section 3.1, the authors illustrates that naively solving an approximate NE of parameterized zero-sum games does not necessarily results in the approximate NE of the original rank-1 games.
- In Section 3.2, the authors sequentially connects the approximate NE of rank-1 games to (constrained) approximate NE of parameterized zero-sum games, and (constrained) approximate NE of the parameterized zero-sum-games to the approximate stationary points of an energy function.
- Finally, the authors propose an algorithm that combines binary search and OMWU that provably learns the approximate NE of the rank-1 game.

**Strengths:**

- The related works are well-studied and well-presented
- Theoretical claims are well-backed with easy-to-follow proofs

**Weaknesses:**

# Technical Novelty
It seems most of the technical heavy-lifting (the reparameterization approach, Lemma 3.3, and the usage of binary search) builds upon  Adsul et al., 2021, and lacks original technical contribution & viewpoint that could be useful for the other researchers in the future.

# Broader Context
I think it would better if the authors could put their theoretical results in a broader context of current literature, and explain why this rank-1 game is an important subclass of differential games.

**Questions:**

# Technical Novelty
I would appreciate if the authors could clarify the the main technical novelties of this work over Adsul et al., 2021 that could contribute (either directly or indirectly) to the future work in this field (differential games, robust optimization, multi-agent RL, etc).

---

> ### Author Response · Authors · 2023-11-19
> **Response to Reviewer cjaZ**
>
> We thank the reviewer for their thoughtful feedback. We appreciate the time and effort they dedicated to reviewing our work.
>
> Below, we address their main concern starting with describing the technical novelty of our work. As emphasized throughout our paper, we acknowledge that the structural properties, particularly that of rank reduction step in Adsul et al. [2018], is also exploited in our approach. However, it is essential to note that the concept of characterizing the strategical equivalence of games dates back to the work of Moulin and Vial [1978] while in the context of rank-1 games, Theobald [2007] introduced a similar parameterization for the case of a quadratic program.
> Combining the parameterization technique with the structural insights from Adler and Monteiro [1992], Adsul et al. [2018] define a polynomial-time algorithm for solving rank-1 games. It is important to highlight that the similarities between our approach and that of Adsul et al. [2018] are confined to that specific aspect.
>
> In contrast, our work takes a distinct path. Our primary focus lies in learning Nash equilibria, a task that inherently does not assume full knowledge of the underlying game; this introduces challenges, and necessitates new techniques. The rank reduction step we mentioned above is the starting point of our approach. Ideally, one might expect that this step alone is sufficient for learning an \textit{approximate} Nash equilibrium in a rank-1 game. However, as detailed in Section 3.1, conventional techniques prove insufficient. Example 3.1 demonstrates that even with knowledge of $\lambda$ (corresponding to a Nash equilibrium of the rank-1 game), naive application of existing methods fail. To address this limitation, in Lemma 3.3 we establish a relationship between the approximate Nash equilibrium of the rank-1 game and those in a parameterized zero-sum, a connection not previously known. In contrary, a similar relationship in Adsul et al. [2018] can be taken for granted in their setting by the rank reduction step.
>
> Leveraging the insights from Lemma 3.3, we introduce the function $f_{\lambda}$, and, as proven in Lemma 3.4, the stationary points of this function can reliably be extended to Nash equilibrium points of the parameterized zero-sum game and consequently to those of rank-1 games. This novel approach provides a fresh perspective on handling rank games, and we anticipate its utility in addressing games of higher rank. Notably, this step is non-trivial and holds true only if the guessed value of $\lambda$ is correct. In contrast, their approach in Adsul et al. [2018] is solely based on the characterization of the optimal faces of the best response polytopes as proven in  Adler and Monteiro [1992].
> As such, the approach in Adsul et al. [2018] requires solving up to four linear programs$^1$ for each value of $\lambda$, whose running time, despite polynomial in the description of the program, is considerably more.
>
> Finally, we elaborate on the last step involving the correct update of the value of $\lambda$. Unlike the approach in Adsul et al. [2018], which involves defining and solving a cumbersome linear program, we propose a simpler rule. As indicated in Lemma 3.5, we can immediately infer the interval containing the correct value of $\lambda$ using only the convexity of Nash equilibria in zero-sum games. The use of binary search, instead of merely increasing or decreasing $\lambda$ by $\epsilon$, is employed to expedite convergence—an elementary strategy when searching within an interval.
>
> In conclusion, our approach offers a more natural and intuitive method for learning rank-1 games. The decentralized algorithm we propose simplifies the analysis significantly, providing a clearer and more straightforward methodology.
>
> From a theoretical standpoint, learning Nash equilibria in bimatrix games beyond zero-sum is a crucial. As detailed in Section 3.1, existing methods for monotone settings (convex-concave optimization) and beyond (including those satisfying (weak) Minty), prove ineffective in our context. Additionally, the independent significance of the optimization problem in Lemma 3.4 holds promise for applications in addressing higher-rank games, a critical open problem, especially for rank-2 games Daskalakis [2018].
>
> $^1$ The first LP finds a primal solution for a particular value of $\lambda$, the second identifies the optimal face of the best response polytope and the rest update the value of $\lambda$.
>
> 1. B. Adsul, J. Garg, R. Mehta, M. Sohoni, and B. V. Stengel. Fast algorithms for rank-1 bimatrix games.
>
> 2. I. Adler and R. D. Monteiro. A geometric view of parametric linear programming.
>
> 3. T. Theobald. Enumerating the nash equilibria of rank 1-games
>
> 4. C. Daskalakis. Equilibria, fixed points, and computational complexity-nevanlinna prize lecture.
>
> 5. H. Moulin and J. P. Vial. Strategically zero-sum games: the class of games whose completely mixed equilibria cannot be improved upon.

---

> > ### Author Response · Authors · 2023-11-21
> > **Have the concerns raised by the reviewer been sufficiently addressed?**
> >
> > We thank again the reviewer for the helpful feedback. Given that the discussion period is soon coming at an end, please let us know if we have adequately addressed the concerns raised, and if the reviewer has any further questions.

---

### Author Response · Authors · 2023-11-19
**Experimental evaluation**

General comment to all reviewers: In response to Reviewer LH17's suggestion for an experimental evaluation, we share a link to an anonymous repo [(link)](https://anonymous.4open.science/api/repo/iclr_rebuttal-B5E7/file/index.html) comparing our method with optimistic gradient ascent over four random rank-1 games. This additional experimental evaluation will be included in a separate section in the appendix. We appreciate the valuable feedback from all reviewers and will continue to make appropriate corrections to further improve the quality of our work.

---

> ### Author Response · Authors · 2023-11-21
> **Rebuttal Revision**
>
> We thank the reviewer for their thoughtful feedback and we appreciate the time and effort they dedicated to reviewing our work.
>
> A revised rebuttal version has been uploaded, with all modifications indicated in blue. We have made thorough efforts to incorporate and address each reviewer's concerns and comments, aiming to further improve the overall quality of our work.

---

### Meta-Review · Area_Chair_TD5k · 2023-12-09

**Metareview:**

The paper is the first to propose gradient based decentralized algorithms for approximate Nash equilibria of rank one games. Reviewers appreciate the novelty of the work and I share the same view and believe that the work makes a nice contribution to the field of learning in games.

**Justification For Why Not Higher Score:**

There is still a gap to lower bound, which is a slightly unsatisfying aspect of the paper.

**Justification For Why Not Lower Score:**

N/A

---

### Decision · Program_Chairs · 2024-01-16

Accept (poster)